# RA-Det: Towards Universal Detection of AI-Generated Images via Robustness Asymmetry

Xinchang Wang [* 1]  Yunhao Chen [* 2]  Yuechen Zhang [1]  Congcong Bian [1]  Zihao Guo [1]  Xingjun Ma [2]  Hui Li [1]

## Abstract

Recent image generators produce photo-realistic content that undermines the reliability of downstream recognition systems. As visual appearance cues become less pronounced, appearance-driven detectors that rely on forensic cues or high-level representations lose stability. This motivates a shift from appearance to behavior, focusing on how images respond to controlled perturbations rather than how they look. In this work, we identify a simple and universal behavioral signal. Natural images preserve stable semantic representations under small, structured perturbations, whereas generated images exhibit markedly larger feature drift. We refer to this phenomenon as **robustness asymmetry** and provide a theoretical analysis that establishes a lower bound connecting this asymmetry to memorization tendencies in generative models, explaining its prevalence across architectures. Building on this insight, we introduce Robustness Asymmetry Detection (RA-Det), a behavior-driven detection framework that converts robustness asymmetry into a reliable decision signal. Evaluated across 14 diverse generative models and against more than 10 strong detectors, RA-Det achieves superior performance, improving the average performance by 7.81%. The method is data- and model-agnostic, requires no generator fingerprints, and transfers across unseen generators. Together, these results indicate that robustness asymmetry is a stable, general cue for synthetic-image detection and that carefully designed probing can turn this cue into a practical, universal detector. The source code is publicly available at GitHub.

---

[*]Equal contribution  [1]Jiangnan University [2]Fudan University. Correspondence to: Hui Li <lihui.cv@jiangnan.edu.cn>.

*Proceedings of the 43$^{rd}$ International Conference on Machine Learning*, Seoul, South Korea. PMLR 306, 2026. Copyright 2026 by the author(s).

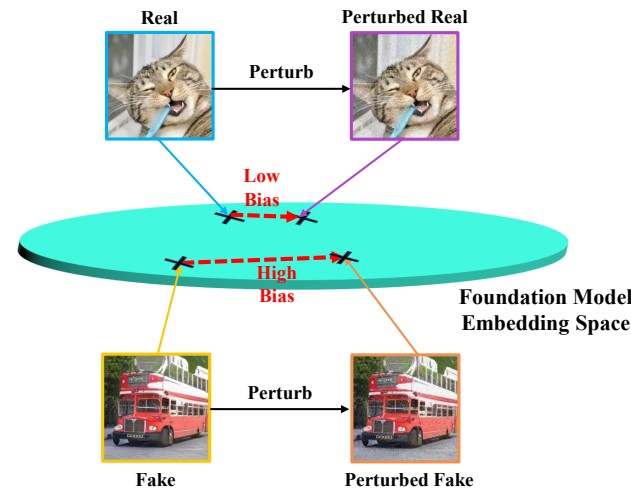

*Figure 1.* Robustness asymmetry between natural and synthetic images in embedding spaces (CLIP (Radford et al., 2021), DINO (Siméoni et al., 2025)). Under perturbation, show minimal embedding displacement, while synthetic images exhibit large representation drift (high displacement).

## 1. Introduction

In recent years, generative models have evolved from early experimental frameworks into mature systems capable of producing highly realistic and often indistinguishable generated images. These technologies, alongside user-friendly platforms (Karras et al., 2019; Rombach et al., 2022; Saharia et al., 2022; Ramesh et al., 2022; Midjourney, 2023; Runway, 2024), empower users to create hyper-realistic content with ease, bringing about the phenomenon where *seeing is no longer believing* (Nightingale & Farid, 2022).

However, the diversity and convenience of generative model technologies, coupled with their ability to generate highly realistic or generated images, also introduce potential risks of misuse and malicious applications(Wang et al., 2024; Lin et al., 2024), such as generating fake information, forging identities and disseminating inappropriate content. Thus, detecting generated images becomes increasingly critical.

Existing generated detection approaches can be broadly grouped into two major paradigms. Representation-driven approaches leverage general-purpose embeddings from

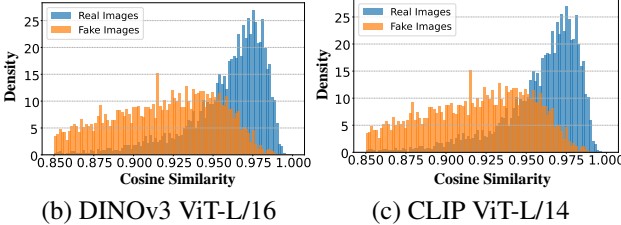

(b) DINOv3 ViT-L/16      (c) CLIP ViT-L/14

*Figure 2.* Backbone-agnostic evidence of robustness asymmetry. We measure embedding stability as the cosine similarity between features extracted from the clean image and its perturbed counterpart. Real images maintain higher similarity than synthetic images, and the separation remains consistent for both DINOv3 and CLIP.

large pre-trained vision models such as CLIP (Radford et al., 2021), DINOv2(Oquab et al., 2024), or vision–language models(Girdhar et al., 2023). These detectors adapt high-level representations via linear probes, adapters, or multi-modal alignment objectives (Ojha et al., 2023; Tan et al., 2025; Yu et al., 2025). Their strength lies in the semantic coverage and generalization ability inherited from the backbone. Artifact-driven forensic methods, in contrast, target generator-induced statistical traces including frequency anomalies(Frank et al., 2020), decoding residuals(Binh & Woo, 2022), upsampling artifacts or local pixel structural disruptions(Tan et al., 2024; Liang et al., 2026). These approaches are lightweight and interpretable, excelling especially when generative pipelines introduce stable structural cues.

Although these two paradigms differ in methodology, they share a common assumption: the decision signal resides in visible appearance, either as explicit low-level artifacts or implicit high-level cues encoded in representations. As generative models reduce classical artifacts and produce distributions closer to natural images, appearance-driven cues grow increasingly subtle and fragile especially under common transformations such as JPEG compression or blur.

In this work, we explore a complementary perspective: instead of analyzing how an image *looks*, we examine how it *behaves* under controlled perturbations. Through extensive studies, we uncover an interesting phenomenon: natural images maintain consistent deep representations when subjected to mild perturbations. Generated images, however, display disproportionately large feature drift. We term this universal generative deficiency as **robustness asymmetry**, as shown in Fig. 1. We futher quantify this effect and Fig. 2 shows that the separation persists across both DINOv3 and CLIP backbones. Moreover, we theoretically derive a lower bound for the robustness asymmetry, showing it is directly linked to the memorization behavior of generative models. This link explains why robustness asymmetry is a widespread phenomenon, as memorization is prevalent across various types of generative models (Carlini et al., 2021; 2023; Chen et al., 2026; 2020a).

Motivated by this insight, we propose Robustness Asymmetry Detection (RA-Det), a behavior-driven framework that operationalizes robustness asymmetry into an effective and generalizable detection signal. RA-Det actively probes image stability via small, semantics-preserving perturbations and integrates complementary evidence through a unified multi-branch architecture. This design captures the full spectrum of generative inconsistencies: it leverages foundation priors for semantic context, explicitly quantifies feature-space displacement via covariance-aware and vector discrepancy metrics, and recovers high-frequency artifacts through a specialized low-level residual stream. By jointly optimizing these diverse signals, RA-Det leverages Robustness Asymmetry that persists across architectures and common post-processing operations.

In short, the main contributions of our proposed framework are summarized as follows:

- We uncover a simple and universal signal–robustness asymmetry between natural and synthetic images, showing that real images remain semantically stable under perturbations, whereas synthetic images demonstrate significantly larger feature-space displacement.

- Based on this insight, we propose RA-Det, a multi-branch detector that aggregates complementary semantic, discrepancy, and low-level residual cues to systematically probe and amplify the robustness discrepancy.

- Following a widely adopted benchmark, our proposed RA-Det outperforms existing detection methods, delivering excellent performance in the generated detection task and fully demonstrating its superior generalization and robustness.

## 2. Related Works

Existing detection methods can be grouped into two paradigms: representation-driven and artifact-driven.

### 2.1. Representation-Driven Detection Approaches

Methods such as UniFD (Ojha et al., 2023) classify CLIP embeddings directly, demonstrating strong generalization from the semantic and structural priors encoded in the backbone. Subsequent works enhance this paradigm by adapting pre-trained models to better capture subtle generative inconsistencies. C2P-CLIP (Tan et al., 2025) introduces cross-view prompting and multi-perspective consistency, LASTED (Wu et al., 2025) integrates textual supervision to refine multimodal representations, and LVLM-based detectors (Yu et al., 2025) leverage large vision–language models for explainable detection. These approaches benefit from broad domain coverage and the strong invariances of foundation models, enabling them to detect synthetic images

beyond specific generative architectures.

## 2.2. Artifact-Driven Forensic Approaches

Frequency-based detectors (Frank et al., 2020) expose anomalies in DCT or Fourier spectra, while gradient-based methods such as LGrad (Tan et al., 2023) capture inconsistencies in local edge structures. Recent works have revisited upsampling and decoding artifacts: NPR (Tan et al., 2024) and FerretNet (Liang et al., 2026) model local pixel relationships and decoder-induced correlations. Diffusion-based reconstruction detectors such as DIRE (Wang et al., 2023) and DRCT (Chen et al., 2024) analyze reconstruction errors to reveal inconsistencies in synthetic content. These approaches are lightweight and interpretable, but their performance often depends on the persistence of structural artifacts, which modern diffusion and rectified-flow models increasingly suppress.

**From Appearance to Behavior** Although representation-driven and artifact-driven methods exploit appearance-level cues, their effectiveness declines as modern generators reduce artifacts and better mimic natural statistics. We shift the focus to **how an image behaves** under structured perturbations. This behavioral view reveals a robust asymmetry between real and synthetic images and provides the foundation for our framework.

## 3. Methodology

In this section, we detail RA-Det, a behavior-driven detection framework, as illustrated in Figure 3.

### 3.1. Overview and Notation

RA-Det is a behavior-driven detection framework built on robustness asymmetry in a frozen foundation feature space (Figure 3). Given an input image $x$ with label $y \in \{0, 1\}$, a frozen encoder $f(\cdot)$ extracts the clean embedding $e = f(x)$. A learnable probing module $\mathcal{P}(\cdot)$ generates a bounded perturbation $\delta$, producing a perturbed view $x' = x + \delta$ and the corresponding embedding $e' = f(x')$. Discrepancy features are derived from $(e, e')$, and a multi-branch detector aggregates complementary evidence from the semantic embedding, discrepancy cues, and low-level residual features to output the final prediction.

### 3.2. Core components

#### 3.2.1. DIFFERENTIAL ROBUSTNESS PROBING

Motivated by the observed robustness asymmetry, we propose a novel probing component named Differential Robustness Probing (DRP) that leverages the differential robustness asymmetry for detection purposes. The primary objective of the DRP is not to attack or fool a detector, but rather to learn and apply a controlled perturbation that maximally amplifies the observable robustness discrepancy between real and fake images.

Architecturally, DRP is implemented as a conditional UNet (Ronneberger et al., 2015) that predicts a pixel-wise perturbation map. The encoder follows a multi-stage downsampling design to extract multi-scale image features, while the clean embedding is projected into spatial feature maps to provide semantic conditioning. At the bottleneck, a cross-attention module integrates the embedding-conditioned features with the image features. The decoder mirrors the encoder with progressive upsampling and skip connections, recovering spatial details at each scale. Finally, a lightweight output head produces the perturbation map, and a `tanh` activation followed by scaling with $\epsilon$ constrains its magnitude, yielding a bounded perturbation for constructing the perturbed image.

Given an input image $x$, we apply designed DRP module $\mathcal{P}(\cdot)$ to obtain the perturbed counterpart $x'$:

$$x' = \mathcal{P}(x) = x + \delta \qquad (1)$$

#### 3.2.2. DISCREPANCY FEATURES IN FOUNDATION SPACE

Both $x$ and $x'$ are projected into the foundation space via $f(\cdot)$, yielding $e = f(x)$ and $e' = f(x')$. RA-Det models robustness asymmetry through discrepancy features derived from $(e, e')$.

**Distance, difference, and similarity.** We compute the embedding distance $d = \|e - e'\|_2$ and the embedding difference vector $\Delta = e - e'$. In addition, the per-sample similarity is defined as

$$s = \cos(e, e'). \qquad (2)$$

**Diagonal-restricted covariance approximation (optional).** To quantify feature displacement with reduced complexity, we further consider a diagonal-restricted covariance approximation (DCA):

$$\text{DCA}(e, e') = \frac{1}{D} \sum_{j=1}^{D} (\Sigma_{ee'})_{jj}, \qquad (3)$$

$$\Sigma_{ee'} = \mathbb{E}[(e - \mathbb{E}[e])(e' - \mathbb{E}[e'])^T], \qquad (4)$$

where $D$ is the embedding dimension and $(\Sigma_{ee'})_{jj}$ denotes diagonal elements. This approximation keeps diagonal terms and discards off-diagonal terms, which show negligible contribution in our ablations.

#### 3.2.3. MULTI-BRANCH DETECTOR AND FUSION

Based on the robustness asymmetry observation, fake images differ from real images not only in high-level semantics(Ojha et al., 2023), but also in how their representations

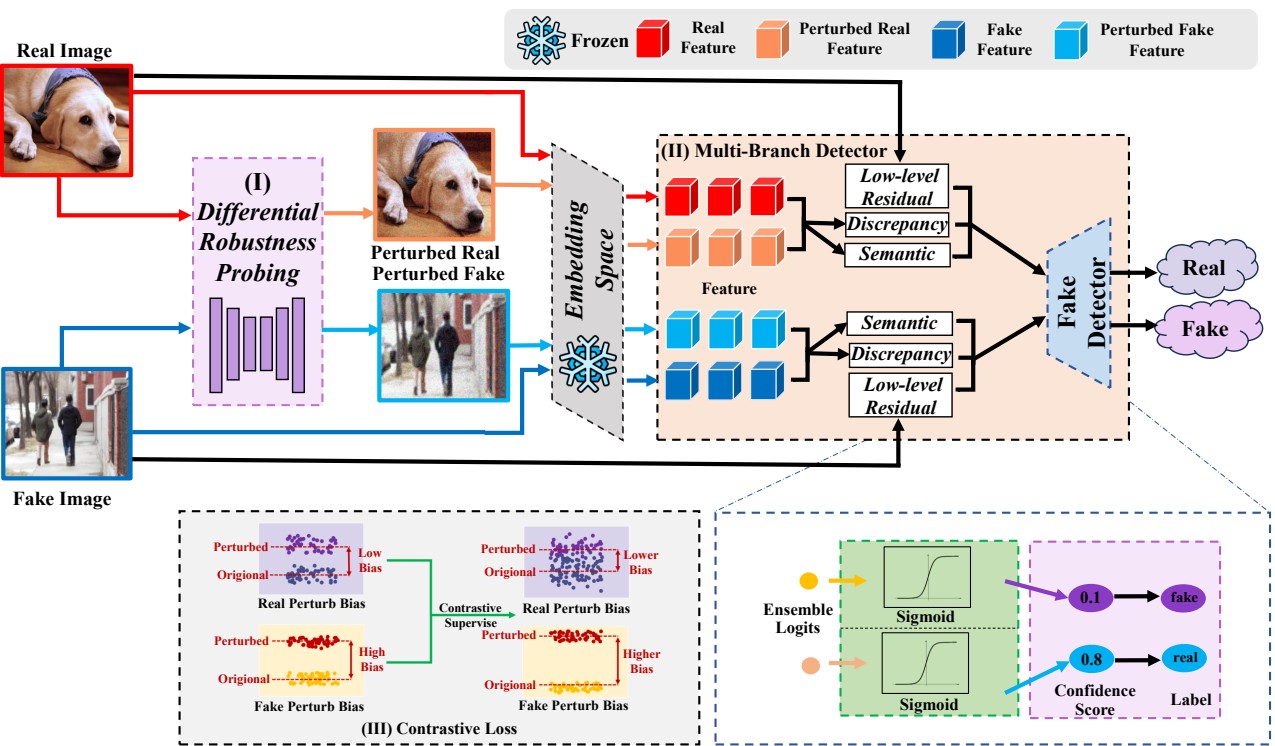

*Figure 3.* **Overview of RA-Det.** The framework processes real and fake images through three key phases to identify generative content: **(I) Differential Robustness Probing (DRP)** generates targeted perturbations to amplify the robustness asymmetry between real and synthetic images. **(II) The Multi-Branch Detector** aggregates complementary forensic evidence by analyzing high-level semantic features, quantifying feature-space stability via the discrepancy branch, and capturing pixel-level artifacts through the low-level residual branch. **(III) Contrastive Loss** guides the training by enforcing a margin between the feature displacements of real and fake images.

change under small, meaning-preserving perturbations, and in subtle low-level artifacts that can be hard to notice visually. RA-Det is therefore built as a multi-branch detector that gathers these complementary cues—semantic evidence, feature discrepancies, and pixel-level residual traces—and aggregates them for the final decision.

**Semantic branch:** The frozen foundation encoder extracts an embedding from the input image, and a lightweight classification head maps this embedding to a logit, providing semantic evidence for real vs. fake.

**Discrepancy branches:** Two additional branches capture how stable the embedding is under perturbation: one MLP takes the DCA distance $d = \|e - e'\|_2$ between clean and perturbed embeddings, and another MLP takes the vector difference $\Delta = e - e'$, so the detector can model both the amount and the pattern of change.

**Low-level residual branch:** Because the foundation encoder emphasizes high-level semantics and layout, RA-Det also uses a pixel-space residual stream to capture subtle synthesis traces. A median filter with a kernel size of 3 produces $\tilde{x}$, the residual map is computed as $r = x - \tilde{x}$, and a lightweight CNN trained from scratch takes $r$ as input to

output a complementary logit. All branch logits are then aggregated to produce the final prediction.

### 3.3. Training Objective

RA-Det is trained with a composite objective:

$$\mathcal{L}_{\text{comp}} = \mathcal{L}_{\text{bce}} + \mathcal{L}_{\text{ra}}, \tag{5}$$

where $\mathcal{L}_{\text{bce}}$ is the standard binary cross-entropy loss, and $\mathcal{L}_{\text{ra}}$ is a hinge-style contrastive loss that separates the similarity statistics of real and fake samples under perturbations.

Given a sample $x_i$ with a perturbed view $x_i' = x_i + \delta_i$, we extract embeddings $e_i = f(x_i)$ and $e_i' = f(x_i')$, and compute the cosine similarity

$$s_i = \cos(e_i, e_i'). \tag{6}$$

For a mini-batch $B$, we split indices into real and fake subsets, $B_{\text{real}} = \{i \in B \mid y_i = 1\}$ and $B_{\text{fake}} = \{i \in B \mid y_i = 0\}$, and define their mean similarities as

$$s_{\text{real}} = \mathbb{E}_{i \in B_{\text{real}}}[s_i], \qquad s_{\text{fake}} = \mathbb{E}_{i \in B_{\text{fake}}}[s_i]. \tag{7}$$

We then enforce a margin $\gamma$ between these batch-level statistics via

$$\mathcal{L}_{\text{ra}} = \text{ReLU}((s_{\text{fake}} - s_{\text{real}}) + \gamma), \tag{8}$$

which penalizes violations of the desired inequality $s_{\text{real}} \geq s_{\text{fake}} + \gamma$ and becomes zero once the margin is satisfied. Unless otherwise stated, we set $\gamma = 0.1$ in all experiments.

# 4. Lower-Bound for the Robustness Asymmetry

In this section, we provide a theoretical lower bound of the robustness gap between real and fake images under our feature–shift operator. We start from the SIDE memorization divergence (Chen et al., 2026), which quantifies the memorization behavior of a generative model (the lower the divergence, the more the generative model memorizes). Building on this, we show that the expected feature shift induced by a small, isotropic perturbation is tightly linked to the memorization of the generative models around training examples. This leads to a lower bound on the robustness difference between real and fake images.

**Notation.** Let $\mathcal{X} \subset \mathbb{R}^n$ be the input domain; $p$ the distribution of real images; $p_\theta$ the distribution of model-generated images; and $D = \{x_i\}_{i=1}^N \subset \mathcal{X}$ the training set. A *fixed $C^2$ encoder* is a feature map $f : \mathcal{X} \to \mathbb{R}^d$ whose parameters are held constant (no further training) and that is twice continuously differentiable with respect to the input. Write $J_f(x) \in \mathbb{R}^{d \times n}$ for the Jacobian of $f$ at $x$ and define the Jacobian energy $G(x) := \|J_f(x)\|_F^2$. Assume $G$ is bounded: $0 \leq G(x) \leq B$ on $\mathcal{X}$. For a small, isotropic probe $\delta \in \mathbb{R}^n$ with $\mathbb{E}[\delta] = 0$ and $\mathbb{E}[\delta\delta^\top] = (\varepsilon^2/n)I_n$, define the feature-shift

$$\text{Shift}_\varepsilon(x) := \mathbb{E}_\delta\left[\|f(x+\delta) - f(x)\|_2^2\right]. \tag{9}$$

All expectations $\mathbb{E}_\mu[\cdot]$ below are with respect to $x \sim \mu$ unless stated otherwise.

**Memorization divergence from SIDE (Chen et al., 2026).** Fix a radius $\varepsilon_0 > 0$ and define the training-neighborhood mixture

$$q_{\varepsilon_0}(x) := \frac{1}{N} \sum_{i=1}^N \mathcal{N}\left(x \mid x_i, \varepsilon_0^2 I_n\right). \tag{10}$$

The SIDE memorization divergence is

$$M(D; p_\theta, \varepsilon_0) := D_{\text{KL}}(q_{\varepsilon_0} \| p_\theta), \tag{11}$$

which is smaller when the generative model memorizes more training data. Define

$$\Delta := \mathbb{E}_{x \sim q_{\varepsilon_0}}[G(x)] - \mathbb{E}_{x \sim p}[G(x)]. \tag{12}$$

Here $\Delta > 0$ expresses that the encoder's Jacobian energy is higher around training neighborhoods than under the real-data distribution.

**Assumption** Following established and recent theories/evidence on augmentation-induced invariance and architectural equivariance in encoders (Chen et al., 2020b; Hounie et al., 2023; Oquab et al., 2024; Rojas-Gomez et al., 2024; Xu et al., 2023; Caron et al., 2021), we make Assumption 4.1.

**Assumption 4.1.** The data lie (locally) on an $m$-dimensional manifold $\mathcal{M} \subset \mathbb{R}^n$ and that a fixed $C^2$ encoder $f$ exhibits reduced local sensitivity along tangent directions and comparatively larger sensitivity along normal directions.

**Rationale.** This assumption reflects the widely used manifold view of natural images, where high-dimensional observations concentrate near a lower-dimensional submanifold of $(\mathbb{R}^n)$. Data augmentation and tangent-based regularization encourage invariance along typical on-manifold transformations (Hounie et al., 2023), which manifests as smaller Jacobian norms in tangent directions, while off-manifold perturbations are not similarly regularized. Architectural constraints further support this pattern: group-equivariant networks explicitly tie representation changes to symmetry actions, reducing sensitivity along symmetry-induced tangents (Chen et al., 2020b; Rojas-Gomez et al., 2024; Xu et al., 2023). Finally, modern self-supervised encoders (e.g., DINO/DINOv2) empirically yield robust, transferable features consistent with augmentation-induced invariance, providing contemporary evidence for the anisotropy posited here (Oquab et al., 2024; Caron et al., 2021). Based on this assumption, we derive the following Lemma 4.2.

**Lemma 4.2** (Small-radius positive margin). *Under the assumption above, there exists $\bar{\varepsilon}_0 > 0$ and a constant $c_0 > 0$ (depending only on the encoder's anisotropy margin and local tube size) such that for all $\varepsilon_0 \in (0, \bar{\varepsilon}_0]$,*

$$\Delta \geq c_0 > 0.$$

We prove Lemma 4.2 to ground the detector's core premise in theory: modern encoders are trained/built to be insensitive along natural, on-manifold transformations but more sensitive off-manifold; thus a small Gaussian "tube" around training samples puts probability mass where the encoder's Jacobian energy is higher, making $\mathbb{E}_{q_{\varepsilon_0}}[G] > \mathbb{E}_p[G]$ and yielding a strictly positive margin ($\Delta > 0$) that drives the robustness gap used in Theorem 1.

**Theorem 4.3** (Lower bound on the shift gap). *Under the setup and Lemma 4.2 above, for sufficiently small $\varepsilon > 0$,*

$$\mathbb{E}_{x \sim p_\theta}\left[\text{Shift}_\varepsilon(x)\right] - \mathbb{E}_{x \sim p}\left[\text{Shift}_\varepsilon(x)\right] \geq \frac{\varepsilon^2}{n}\left(\Delta - B\sqrt{M(D; p_\theta, \varepsilon_0)/2}\right) + O(\varepsilon^4). \tag{13}$$

More specifically, Theorem 4.3 shows that the expected shift under $p_\theta$ is lower-bounded by a term that decreases

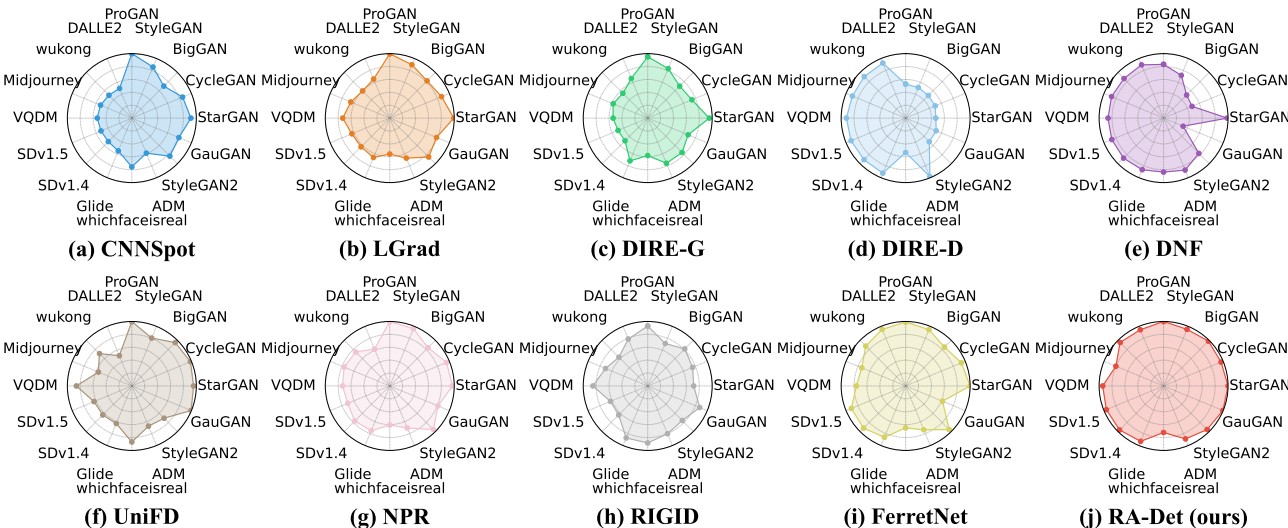

*Figure 4.* Radar chart comparison of representative detectors across 16 generators. We compare GAN-specific detectors, including CNNSpot (Wang et al., 2020) and LGrad (Tan et al., 2023); diffusion-specific detectors, including DIRE-G and DIRE-D (Wang et al., 2023) and DNF (Zhang & Xu, 2025); and universal detectors, including UniFD (Ojha et al., 2023), NPR (Tan et al., 2024), RIGID (He et al., 2024), FerretNet (Liang et al., 2026), and RA-Det. Each subplot shows the per-generator accuracy of one detector, where a larger and more balanced radar profile indicates stronger cross-generator generalization.

with the SIDE divergence $M(D; p_\theta, \varepsilon_0)$. In other words, more memorization of the training data provably creates a larger robustness asymmetry in the small-noise regime. Prior works on extraction and membership inference in different generative models demonstrate that such memorization is prevalent across GANs, diffusion models, and even the large language models (Hayes et al., 2019; Chen et al., 2020a; Hilprecht et al., 2019; Carlini et al., 2023; 2021). This explains why the robustness asymmetry is *universal* because memorization is prevalent across a wide range of generative models types. The proof and empirical validation of the lower bound are in Appendix D.

## 5. Experiments

### 5.1. Experimental Setup

#### 5.1.1. DATASETS

To better simulate real-world black-box detection scenarios, we follow the protocol established by Wang et al. restricting training to one generative model while evaluating generalization on unseen models. Specifically, during the training phase, we use only 360k real images and 360k synthetic images generated by ProGAN (Karras et al., 2018). The evaluation phase, however, is conducted on ProGAN and 15 unseen generative models or platforms provided by Zhong et al. (2024). The test set includes both facial images and other semantic categories, covering generative models and platforms such as GAN-based models, diffusion-based models, and platform-generated content, ensuring a robust and diverse assessment of the model's generalization ability.

In addition to the established benchmark generators, we further evaluate RA-Det on more recent synthesis paradigms, including recent diffusion and platform generators such as SDXL (Podell et al., 2024), FLUX.1 (Labs, 2024), and Midjourney v6 (Midjourney Team, 2024), as well as autoregressive generators such as Infinity (Han et al., 2025) and VAR (Tian et al., 2024). The full results are reported in Appendix A.

#### 5.1.2. IMPLEMENTATION DETAILS

RA-Det is implemented in PyTorch using DINOv3 ViT-L/16 as the fixed backbone. All images are resized to $256 \times 256$ and randomly cropped to $224 \times 224$ during training, while center cropping is applied at testing. We train the perturbation generator and detection head for 10 epochs using the Adam optimizer with an initial learning rate of $1 \times 10^{-5}$ and a batch size of 32. JPEG compression and blur augmentations are disabled to ensure that robustness asymmetry arises solely from the probing module. All experiments are conducted on eight NVIDIA H200 GPUs.

#### 5.1.3. COMPARED METHODS

We classify existing methods into three categories: **GAN-specific detectors**, such as FreDect (Frank et al., 2020) and LGrad (Tan et al., 2023); **Diffusion-specific detectors**, including DIRE (Wang et al., 2023) and DNF (Zhang & Xu, 2025); and **Universal detectors**, like CNNSpot (Wang et al., 2020), GramNet (Liu et al., 2020), UniFD (Ojha et al., 2023), NPR (Tan et al., 2024), RIGID (He et al., 2024) and FerretNet (Liang et al., 2026).

*Table 1.* The detection performance comparison between our approach and baselines. We use ACC(%)/AP(%) as the evaluation metrics. DIRE-D denotes the DIRE detector trained over fake images from ADM, while DNF uses a model trained on ADM, similar to DIRE-D. DIRE-G denotes the DIRE detector trained over the same training set (ProGAN) as others. Among all detectors, the best result and the second-best result are denoted in boldface and underline, respectively.

| Generator | GAN Detector | | | Diffusion Detector | | Universal Detector | | | | | | |
| --- | --- | --- | --- | --- | --- | --- | --- | --- | --- | --- | --- | --- |
| | FreDect | LGrad | DIRE-G | DIRE-D | DNF | CNNSpot | GramNet | UniFD | NPR | RIGID | FerretNet | RA-Det (ours) |
| ProGAN | 99.36/_99.99_ | 99.78/_99.99_ | 95.19/99.08 | 52.75/58.79 | 83.35/60.40 | **99.99/100.00** | _99.99/100.00_ | 99.81/**100.00** | 99.90/99.98 | 93.00/96.83 | 99.09/99.95 | _99.98/100.00_ |
| StyleGAN | 78.02/88.98 | 89.63/98.03 | 83.03/91.74 | 51.31/56.68 | 71.76/44.18 | 85.71/_99.54_ | 83.59/94.49 | 80.40/97.48 | **96.06/99.78** | 71.19/70.61 | 94.53/98.63 | _94.98_/98.90 |
| BigGAN | 81.98/93.62 | 81.73/89.08 | 70.12/75.25 | 49.70/46.91 | 50.55/51.75 | 70.19/84.51 | 67.33/81.79 | _95.08/99.27_ | 83.95/85.59 | 81.25/85.69 | 85.18/91.06 | **98.35/99.83** |
| CycleGAN | 78.77/84.78 | 86.94/95.01 | 74.19/80.56 | 49.58/50.03 | 47.77/66.49 | 85.20/93.48 | 86.07/95.33 | **98.33/99.80** | 95.19/98.12 | 75.34/83.04 | 93.64/98.34 | _96.52/99.47_ |
| StarGAN | 94.62/_99.49_ | 99.27/**100.00** | 95.47/99.34 | 46.72/40.64 | 98.55/86.44 | 91.62/98.15 | 95.05/99.23 | 95.75/99.37 | 97.17/**100.00** | 71.34/77.64 | **99.97/100.00** | _99.92/100.00_ |
| GauGAN | 80.56/82.84 | 78.46/95.43 | 67.79/72.15 | 51.23/47.34 | 32.78/39.48 | 78.93/89.49 | 69.35/84.99 | _99.47/99.98_ | 80.94/82.97 | 87.26/93.71 | 61.84/59.71 | **99.51**/_99.77_ |
| StyleGAN2 | 66.19/82.54 | 85.32/97.89 | 75.31/88.29 | 51.72/58.03 | 77.42/40.16 | 83.39/99.05 | 87.28/99.11 | 70.76/97.71 | **95.61/99.95** | 75.78/80.78 | _95.56/99.16_ | 95.54/99.15 |
| whichfaceisreal | 50.75/55.85 | 55.70/57.99 | 58.05/60.13 | 53.30/59.02 | 83.70/52.64 | 75.65/83.11 | 72.70/94.22 | 86.90/**96.73** | 60.60/62.91 | **88.40**/_95.06_ | 64.95/70.36 | 72.20/81.13 |
| ADM | 63.40/61.77 | 67.15/72.95 | 75.78/85.84 | **98.25/99.79** | 87.22/83.72 | 58.78/71.07 | 58.61/73.11 | 67.46/89.80 | 70.58/75.08 | 80.22/85.88 | 73.65/83.23 | _88.89/95.07_ |
| Glide | 54.13/52.91 | 66.11/80.42 | 71.75/78.35 | _92.42/99.54_ | 86.62/95.18 | 55.00/66.16 | 54.50/66.76 | 63.09/83.81 | 75.18/83.24 | 87.26/93.89 | 86.12/93.62 | **92.94**/_97.83_ |
| SDv1.4 | 38.79/37.83 | 63.02/62.37 | 49.74/49.87 | 91.24/_98.61_ | 87.77/79.22 | 51.55/56.88 | 51.70/59.83 | 63.66/86.14 | 76.81/82.79 | 63.57/65.59 | _92.33_/97.20 | **96.50/99.08** |
| SDv1.5 | 39.21/37.76 | 63.67/62.85 | 49.83/49.52 | _91.63/98.83_ | 86.86/77.54 | 52.16/60.37 | 52.86/61.13 | 63.49/85.84 | 73.22/77.74 | 85.01/90.93 | 91.13/96.64 | **96.01/98.92** |
| VQDM | 77.80/85.10 | 72.99/77.47 | 53.68/54.57 | _91.90/98.98_ | 86.15/77.14 | 53.67/61.92 | 52.86/61.13 | 86.01/96.53 | 73.22/77.74 | 85.01/90.93 | 76.44/85.23 | **94.43/98.68** |
| Midjourney | 45.87/46.09 | 65.35/71.86 | 58.01/61.86 | **89.45/97.32** | 87.32/68.09 | 52.59/55.90 | 50.02/56.82 | 56.13/74.00 | 76.61/82.61 | 70.80/78.63 | 73.66/80.27 | _80.04/87.53_ |
| wukong | 40.30/39.58 | 59.55/62.48 | 54.46/55.38 | _90.90/98.37_ | 86.80/87.32 | 50.23/52.85 | 50.76/55.62 | 70.93/91.07 | 74.45/78.17 | 63.09/67.10 | 87.48/94.72 | **95.33/98.33** |
| DALLE2 | 34.67/38.22 | 65.45/82.55 | 66.48/74.48 | 92.45/**99.71** | 89.39/89.35 | 49.82/50.59 | 49.25/49.82 | 50.75/63.04 | 61.88/71.64 | 78.74/87.28 | **94.95**/_98.82_ | _94.30_/98.14 |
| Average | 64.03/68.28 | 75.34/84.35 | 68.68/74.13 | 71.53/75.87 | 77.75/68.69 | 68.38/76.39 | 67.58/76.93 | 78.00/89.73 | 80.55/84.05 | 77.17/82.38 | _85.66/90.43_ | **93.47/97.00** |

### 5.1.4. EVALUATION METRICS

Following the protocol established by PatchCraft (Zhong et al., 2024), we adopt Accuracy (ACC) and Average Precision (AP), two widely used metrics in generative image detection (Wang et al., 2020; Ojha et al., 2023; Tan et al., 2024). These metrics provide a comprehensive evaluation of detection performance. We report both values for each method across all datasets and compute the mean performance to assess generalization.

### 5.2. Comparison of Detection Performance

We evaluate RA-Det under a comprehensive cross-generator setting, covering 16 diverse generative models and more than 10 representative detectors, including GAN-specific, diffusion-specific, and universal approaches. As reported in Table 1, RA-Det achieves the best overall performance, reaching an average accuracy of **93.47%** and an average AP of **97.00%**. This result surpasses the strongest universal baseline, FerretNet (85.66% / 90.43%), by a substantial margin, demonstrating a clear advantage in both accuracy and ranking quality. Fig. 4 provides a complementary per generator view of the same comparison. RA-Det forms a larger and more balanced radar profile, indicating that its advantage does not come from only a few generators but from more stable cross-generator performance.

In contrast, existing detectors exhibit clear limitations in generalization. Appearance-driven or architecture-specific methods often perform well only within their target domains, such as GAN-based or diffusion-based generators, but degrade noticeably when evaluated on unseen or mismatched models. Universal detectors, including CNNSpot, GramNet, and UniFD, improve cross-model robustness to some extent, yet their reliance on static appearance cues or representa-

*Table 2.* Average performance under different training-source settings. Setting I, Setting II, and Setting III correspond to ProGAN-only, SDv1.4-only, and mixed ProGAN+SDv1.4 training, respectively.

| Setting | Training source | Acc (%) | AP (%) |
| --- | --- | --- | --- |
| Setting I | ProGAN | 93.47 | 97.00 |
| Setting II | SDv1.4 | 92.70 | 96.99 |
| Setting III | ProGAN + SDv1.4 | 93.33 | 96.69 |

tion similarity restricts stability under diverse generative processes. NPR and FerretNet, which exploit local pixel dependencies introduced by upsampling operations, achieve competitive performance on several generators, supporting the presence of structured artifacts in synthetic images. However, their effectiveness remains uneven across models, particularly for more challenging or stylistically diverse generators.

Compared with RIGID, a behavior-based baseline that also relies on a robustness gap, RA-Det models robustness asymmetry through controlled perturbations and learns a decision function over the resulting discrepancy features. RA-Det improves the average ACC/AP from 77.17% / 82.38% for RIGID to 93.47% / 97.00%, showing that learned discrepancy modeling is substantially more effective than threshold-based robustness scoring.

### 5.3. Generality of Robustness Asymmetry

The main experiments follow the standard single generator training protocol, where RA-Det is trained with ProGAN fake images. To examine whether robustness asymmetry is a general property of generated images rather than a cue tied to one training generator, we further evaluate RA-Det under three training settings. Setting I uses ProGAN as the fake

training source, Setting II uses SDv1.4, and Setting III uses both ProGAN and SDv1.4. As shown in Table 2, all three settings achieve comparable average performance. This indicates that the learned robustness asymmetry cue remains stable across different training sources and is not specific to ProGAN or GAN-based synthesis. The detailed dataset composition, training-source statistics, and complete results for each individual generator are provided in Appendix A.

## 5.4. Ablation Study

We conduct ablation experiments to quantify the contribution of the frozen foundation backbone, the robustness-aware training objective, the probing strategy, the contrastive margin, and each branch in the multi-branch detector of RA-Det. Table 3 summarizes all variants. For readability, **Sem/Dis/Res** denote the semantic branch, discrepancy branch, and low-level residual branch, respectively, where the discrepancy branch is instantiated by the embedding distance $d$ and difference vector $\Delta$ defined in Sec. 3.2.3.

**Backbone.** To verify that RA-Det is not tied to a specific representation space, we replace the frozen backbone while keeping the detector design and training objective unchanged. As shown in Table 3, RA-Det achieves strong performance with both DINOv3 ViT-L/16 and CLIP ViT-L/14. This indicates that the robustness asymmetry cue is not restricted to a single representation space. The stronger result with DINOv3 further shows that a more discriminative visual representation can better amplify the advantage of our robustness asymmetry modeling.

**Training objective.** We study the role of the robustness-aware objective by training RA-Det with only the binary classification loss. Without $\mathcal{L}_{ra}$, the detector still receives supervision from real/fake labels, but the probing module is no longer explicitly encouraged to separate real and generated images by their representation stability. As a result, the model can rely more on static classification cues and less on the behavioral discrepancy revealed by perturbation. The performance drop in Table 3 shows that $\mathcal{L}_{ra}$ helps align DRP with the core robustness asymmetry signal, making the learned discrepancy features more discriminative.

**Margin sensitivity.** We further evaluate the contrastive margin $\gamma$ in $\mathcal{L}_{ra}$. As shown in Table 3, RA-Det remains stable across a broad range of margin values. The default setting $\gamma = 0.1$ is competitive with the best result, suggesting that the method is not sensitive to this hyperparameter.

**Probing strategy.** We further replace DRP with several simple perturbation operators, including Gaussian noise, Gaussian blur, and resize residuals computed from down-sampling and upsampling differences. As shown in Table 3, all hand-crafted perturbations perform clearly worse than the learned DRP. This suggests that generic perturbations

*Table 3.* Ablation on backbone, training objective, probing strategy, contrastive margin, and the multi-branch detector in Sec. 3.2.3. **Sem/Dis/Res** denote the semantic branch, discrepancy branch, and low-level residual branch, respectively. The full model used in our main experiments is reported at the top, with the best results shown in bold.

| Variant | Backbone | Loss | Acc (%) | AP (%) |
|---|---|---|---|---|
| *Full model used in the main experiments* | | | | |
| **Full RA-Det (Ours)** | DINOv3 ViT-L/16 | $\mathcal{L}_{comp}$ | **93.47** | **97.00** |
| *Backbone / Objective* | | | | |
| Backbone → CLIP | CLIP ViT-L/14 | $\mathcal{L}_{comp}$ | 91.85 | 95.00 |
| Loss → BCE | DINOv3 ViT-L/16 | $\mathcal{L}_{bce}$ | 90.86 | 95.26 |
| *Probing strategy* | | | | |
| DRP → Gaussian noise | DINOv3 ViT-L/16 | $\mathcal{L}_{comp}$ | 86.11 | 91.71 |
| DRP → Gaussian blur | DINOv3 ViT-L/16 | $\mathcal{L}_{comp}$ | 87.07 | 91.42 |
| DRP → Resize residual | DINOv3 ViT-L/16 | $\mathcal{L}_{comp}$ | 86.59 | 90.83 |
| *Contrastive margin* | | | | |
| $\gamma = 0.1$ (default) | DINOv3 ViT-L/16 | $\mathcal{L}_{comp}$ | 93.47 | 97.00 |
| $\gamma = 0.25$ | DINOv3 ViT-L/16 | $\mathcal{L}_{comp}$ | 90.96 | 95.07 |
| $\gamma = 0.5$ | DINOv3 ViT-L/16 | $\mathcal{L}_{comp}$ | 92.67 | 96.23 |
| $\gamma = 1.0$ | DINOv3 ViT-L/16 | $\mathcal{L}_{comp}$ | 93.47 | 97.02 |
| $\gamma = 2.0$ | DINOv3 ViT-L/16 | $\mathcal{L}_{comp}$ | 92.31 | 96.16 |
| *Multi-branch detector* | | | | |
| w/o Sem | DINOv3 ViT-L/16 | $\mathcal{L}_{comp}$ | 90.34 | 95.43 |
| w/o Dis | DINOv3 ViT-L/16 | $\mathcal{L}_{comp}$ | 85.73 | 91.52 |
| w/o Res | DINOv3 ViT-L/16 | $\mathcal{L}_{comp}$ | 88.62 | 94.26 |
| Res: Med → Gau | DINOv3 ViT-L/16 | $\mathcal{L}_{comp}$ | 90.58 | 95.63 |

can reveal limited robustness differences, but they are insufficient to fully expose the robustness asymmetry cue. Thus, the learned probing module is important for amplifying the behavioral gap between real and generated images.

**Multi-branch detector.** We further evaluate the contribution of each branch in Sec. 3.2.3. The discrepancy branch is the most critical component, as removing it leads to the largest performance drop. This agrees with our design, since this branch directly captures the robustness asymmetry cue. The residual branch provides useful low-level forensic cues, while the semantic branch brings only a mild gain. In particular, our low-level residual branch uses a median filter to construct the residual map. To examine this design choice, we replace it with a Gaussian filter based alternative. The Gaussian variant performs worse than the default design but better than removing the residual branch. This indicates that residual cues complement foundation-space features, and that the median-based construction better preserves detection-relevant local traces.

## 5.5. Robustness Testing

In real-world scenarios, generative image detection systems inevitably encounter post-processing operations such as JPEG compression and Gaussian blur during image transmission and storage. Evaluating robustness under such perturbations is therefore essential for practical deployment. We conduct robustness experiments under controlled JPEG compression (QF = 95, 90, 85) and Gaussian blur ($\sigma = 0.8$, 1.0, 1.5), and compare RA-Det with representative artifact-

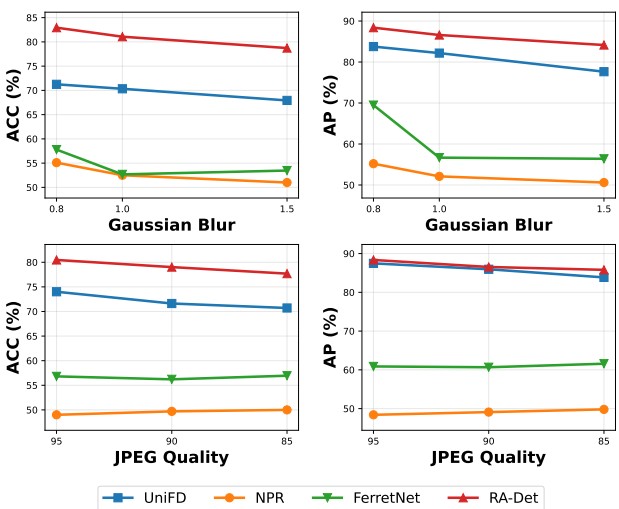

*Figure 5.* Performance comparison under common image perturbations. The plots show detector performance against (a) JPEG compression (QF = 95, 90 and 85) and (b) Gaussian blur ($\sigma$ = 0.8, 1.0 and 1.5). The mean scores (diamonds) indicate that our method, RA-Det, maintains the highest accuracy and AP, demonstrating superior robustness.

driven detectors (NPR and FerretNet) and the representation-driven baseline UniFD.

As shown in Fig. 5, RA-Det consistently achieves the best ACC and AP under both Gaussian blur and JPEG compression. Artifact-driven methods degrade more clearly because these operations can disrupt the local pixel statistics and generation artifacts used for detection. UniFD is more stable than artifact-driven baselines, but it still performs worse than RA-Det across all settings. These results suggest that robustness asymmetry provides a reliable cue under common image degradation, supporting the practical robustness of RA-Det in post-processed scenarios.

### 5.6. Efficiency Analysis

We further analyze the computational cost of RA-Det. RA-Det follows the common practice of using a large pretrained visual backbone for representation extraction, and additionally introduces a conditional UNet based DRP module to expose robustness asymmetry. In our evaluation environment, RA-Det takes 25.14 ms per image, while representative detectors take 16.49 ms and 21.64 ms. The detailed FLOPs, parameters, memory usage, and inference time are reported in Appendix B. These results show that the extra cost mainly comes from active probing on top of foundation features, while the overall runtime remains moderate. Notably, by projecting the clean and perturbed views into the same foundation embedding space, RA-Det jointly derives semantic evidence and robustness-asymmetry cues for the two corresponding branches in Sec. 3.2.3, without introduc-

ing separate feature extractors or requiring extra backbone passes for individual branches. Reducing the probing cost remains an important direction for future work.

## 6. Conclusion

In this work, rather than following previous approaches that focus on identifying universal artifacts of fake images, we propose an innovative method that distinguishes real and generated images based on their inherent differences in robustness against perturbations. Experimental results demonstrate that our method not only surpasses previous artifact-based approaches but also exhibits strong generalization ability across multiple generative models. This validates that leveraging the disparity in robustness helps prevent the detector from merely learning superficial features of the training data while failing to truly distinguish real and generated images. Furthermore, robustness experiments confirm that this assumption remains valid even under image distortions, ensuring that the detection model does not fail due to minor degradations in image quality. In future work, we plan to investigate whether the robustness difference between real and fake images is a general phenomenon that extends to more diverse forms of perturbations, such as adversarial attacks.

## Acknowledgements

This work was supported by the National Key Research and Development Program of China (2023YFE0116300), the National Natural Science Foundation of China (62202205) and the Natural Science Foundation of Xinjiang Uygur Autonomous Region (2025D01A50).

## Impact Statement

This paper studies the detection of AI-generated images to improve trust in visual content and reduce risks from deceptive synthetic media. RA-Det may support content verification, platform moderation, forensic analysis, and the protection of downstream recognition systems by using robustness asymmetry as a complementary signal beyond visual artifacts. At the same time, the method should not be used as a fully automated decision maker in high-stakes scenarios, since false positives may incorrectly flag authentic images and false negatives may miss synthetic ones. RA-Det also introduces extra inference cost due to the foundation backbone and probing module, and the common single-generator training protocol may not fully reflect real-world data distributions. We therefore view this work as part of a broader defensive effort that should be combined with human oversight, continued robustness evaluation, more realistic benchmarks, and complementary provenance mechanisms.

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

## A. Training-Source Analysis for the Universality of Robustness Asymmetry

To examine whether the learned robustness-asymmetry cue depends on a particular training generator, we evaluate RA-Det under three training-source settings, covering GAN-only training, diffusion-only training, and mixed-generator training. We use LSUN (Yu et al., 2015) as the source of real images. For diffusion-generated images, we use the SDv1.4 subset from AIGIBench (Li et al., 2026), whose synthetic images are sourced from GenImage (Zhu et al., 2023). The detailed settings are defined as follows.

**Setting-I.** 360K real images from LSUN and 360K fake images from ProGAN (Karras et al., 2018).

**Setting-II.** 52,922 real images from LSUN and 72,000 fake images from the SDv1.4 subset of AIGIBench.

**Setting-III.** 72,000 real images from LSUN and 72,000 fake images sampled from both ProGAN and the SDv1.4 subset of AIGIBench.

As shown in Table 4, RA-Det achieves stable average performance under all three training-source settings. Specifically, the ProGAN-only, SDv1.4-only, and mixed-generator settings achieve average accuracies of 93.47%, 92.70%, and 93.33%, respectively, with corresponding AP scores of 97.00%, 96.99%, and 96.69%. The small variation across these settings indicates that the learned robustness-asymmetry cue is not specific to the generator used for training DRP.

These results support our hypothesis that robustness asymmetry is a general property of generated images rather than an artifact tied to ProGAN or GAN-based synthesis. Importantly, training DRP with diffusion-generated images still yields strong overall performance, showing that the asymmetry exploited by RA-Det remains valid under diffusion-based training. The per-generator results further suggest that diffusion-source training can improve performance on several diffusion-based and recent generators, supporting the transferability of robustness asymmetry across different synthesis paradigms. Full per-generator results are reported in Table 4.

*Table 4.* Per-generator performance of RA-Det under different training-source settings. Setting I, Setting II, and Setting III correspond to ProGAN-only, SDv1.4-only, and mixed ProGAN+SDv1.4 training, respectively. Generators are grouped into recent generative models and established benchmark generators.

*(a)* Accuracy (ACC, %) across individual generators under each training-source setting. The last column reports the average across generators.

| Setting | Recent Generative Models | | | | | | | Established Benchmark Generators | | | | | | | | | | | | | | | | Avg. |
|---|---|---|---|---|---|---|---|---|---|---|---|---|---|---|---|---|---|---|---|---|---|---|---|---|
| | SDXL | SDv3.5 | FLUX | FLUX1-dev | MJ-v6 | Infinity | VAR | ProGAN | StyleGAN | BigGAN | CycleGAN | StarGAN | GauGAN | StyleGAN2 | WFIR | ADM | Glide | SDv1.4 | SDv1.5 | VQDM | MJ | Wukong | DALLE2 | |
| Setting I | 96.29 | 75.61 | 57.19 | 80.58 | 74.97 | 88.32 | 91.63 | 99.98 | 94.98 | 98.35 | 96.52 | 99.92 | 99.51 | 95.54 | 72.20 | 88.89 | 92.94 | 96.50 | 96.01 | 94.43 | 80.04 | 95.33 | 94.30 | 93.47 |
| Setting II | 94.42 | 91.85 | 85.35 | 90.00 | 81.23 | 97.08 | 79.60 | 95.49 | 96.48 | 90.83 | 82.17 | 95.75 | 91.21 | 96.21 | 86.41 | 78.87 | 95.60 | 98.95 | 98.65 | 93.88 | 94.61 | 97.92 | 91.10 | 92.70 |
| Setting III | 90.58 | 82.35 | 62.72 | 81.21 | 72.83 | 92.03 | 91.65 | 99.39 | 93.32 | 96.10 | 95.68 | 100.00 | 93.10 | 90.23 | 85.10 | 85.32 | 93.17 | 96.81 | 96.99 | 92.32 | 87.07 | 95.18 | 93.95 | 93.33 |

*(b)* Average precision (AP, %) across individual generators under each training-source setting. The last column reports the average across generators.

| Setting | Recent Generative Models | | | | | | | Established Benchmark Generators | | | | | | | | | | | | | | | | Avg. |
|---|---|---|---|---|---|---|---|---|---|---|---|---|---|---|---|---|---|---|---|---|---|---|---|---|
| | SDXL | SDv3.5 | FLUX | FLUX1-dev | MJ-v6 | Infinity | VAR | ProGAN | StyleGAN | BigGAN | CycleGAN | StarGAN | GauGAN | StyleGAN2 | WFIR | ADM | Glide | SDv1.4 | SDv1.5 | VQDM | MJ | Wukong | DALLE2 | |
| Setting I | 91.34 | 81.65 | 61.50 | 88.40 | 70.93 | 80.09 | 84.16 | 100.00 | 98.90 | 99.83 | 99.47 | 100.00 | 99.77 | 99.15 | 81.13 | 95.07 | 97.83 | 99.08 | 98.92 | 98.68 | 87.53 | 98.33 | 98.14 | 97.00 |
| Setting II | 98.95 | 96.14 | 91.01 | 96.44 | 89.96 | 99.51 | 88.75 | 99.00 | 99.27 | 96.44 | 89.55 | 99.27 | 97.32 | 99.37 | 93.28 | 87.76 | 98.34 | 99.83 | 99.83 | 98.29 | 98.11 | 99.72 | 96.10 | 96.99 |
| Setting III | 94.69 | 74.21 | 55.80 | 88.89 | 75.64 | 94.49 | 96.41 | 99.95 | 97.84 | 98.49 | 99.17 | 100.00 | 93.82 | 96.51 | 87.57 | 93.50 | 97.03 | 99.31 | 99.32 | 96.65 | 91.49 | 98.12 | 98.43 | 96.69 |

## B. Efficiency Analysis

To better understand the practical cost of RA-Det, we compare it with representative static detectors that use foundation model features. All methods are evaluated under the same hardware environment and input resolution. As shown in Table 5, RA-Det introduces additional computation due to the DRP module and the second feature extraction pass on the perturbed image. Nevertheless, the inference time remains within a moderate range compared with static foundation feature based detectors.

## C. On DIRE and Dataset–Format Biases

The notably high performance of DIRE on diffusion-generated images is largely enabled by unintended dataset biases rather than purely generator-specific artifacts. DIRE(Wang et al., 2023) is trained on a benchmark where synthetic images are stored in lossless PNG format and often at fixed resolutions, while real photographs are saved in JPEG format with varying compression levels and sizes. In effect, the detector learns to pick up on format cues (for example, JPEG compression

*Table 5.* Computational cost comparison with representative static detectors. All methods are evaluated under the same hardware environment and input resolution.

| Method | FLOPs (G) | Params (M) | Peak Mem. (MB) | Time (ms) |
|---|---|---|---|---|
| UniFD (Ojha et al., 2023) | 103.79 | 202.05 | 1697.95 | 16.49 |
| C2P-CLIP (Tan et al., 2025) | 207.23 | 303.97 | 1835.00 | 21.64 |
| RA-Det (Ours) | 281.48 | 378.12 | 1962.00 | 25.14 |

artifacts or size/resolution uniformity (Grommelt et al., 2024)) rather than genuine generative fingerprints. When these format differences are removed or equalized, its performance degrades substantially and cross-generator transfer drops sharply

## D. Lower-Bound for the Robustness Asymmetry

### D.1. Theoretical Derivation

We derive a lower bound for the robustness gap between real and model-generated images under a small, isotropic probe in feature space. We begin from the SIDE memorization divergence (Chen et al., 2026), which quantifies generator memorization around training samples: smaller divergence indicates stronger memorization. We then connect memorization to the expected feature shift under small perturbations, yielding a quantitative lower bound on the robustness difference.

**Notation.** Let the ambient space be $\mathcal{X} \subset \mathbb{R}^N$; $p$ the real image distribution; $p_\theta$ the model-generated distribution; and $D = \{x_i\}_{i=1}^{N_{\mathrm{tr}}} \subset \mathcal{X}$ the generator's training set. A *fixed $C^2$ encoder* is a twice continuously differentiable map $f : \mathcal{X} \to \mathbb{R}^d$ whose parameters are frozen. Denote the Jacobian by $J_f(x) \in \mathbb{R}^{d \times N}$ and the Jacobian energy by $G(x) := \|J_f(x)\|_F^2$, assumed bounded on $\mathcal{X}$: $0 \leq G(x) \leq B$. For a small, isotropic probe $\boldsymbol{\eta} \in \mathbb{R}^N$ with $\mathbb{E}[\boldsymbol{\eta}] = 0$ and $\mathbb{E}[\boldsymbol{\eta}\boldsymbol{\eta}^\top] = (\varepsilon^2/N)I_N$, define the feature shift

$$\mathrm{Shift}_\varepsilon(x) := \mathbb{E}_{\boldsymbol{\eta}}\big[\|f(x+\boldsymbol{\eta}) - f(x)\|_2^2\big]. \tag{14}$$

All expectations $\mathbb{E}_\mu[\cdot]$ below are with respect to $x \sim \mu$ unless stated otherwise.

**Memorization divergence (SIDE) (Chen et al., 2026; 2025).** Fix a radius $\varepsilon_0 > 0$ and define the training-neighborhood mixture

$$q_{\varepsilon_0}(x) := \frac{1}{N_{\mathrm{tr}}} \sum_{i=1}^{N_{\mathrm{tr}}} \mathcal{N}\Big(x \,\Big|\, x_i, \varepsilon_0^2 I_N\Big). \tag{15}$$

SIDE memorization is

$$M(D; p_\theta, \varepsilon_0) := D_{\mathrm{KL}}(q_{\varepsilon_0} \,\|\, p_\theta), \tag{16}$$

smaller when $p_\theta$ puts more mass near training examples. Define

$$\Delta := \mathbb{E}_{x \sim q_{\varepsilon_0}}[G(x)] - \mathbb{E}_{x \sim p}[G(x)]. \tag{17}$$

**Tangent/normal anisotropy (modeling hypothesis).** Motivated by recent evidence that augmentation and architectural equivariance induce on-manifold invariances in modern encoders (Oquab et al., 2024; Caron et al., 2021; Rojas-Gomez et al., 2024; Xu et al., 2023), we *refer to* Assumption 4.1 (stated elsewhere): natural images lie locally on an $m$-dimensional manifold $\mathcal{M} \subset \mathbb{R}^N$ and a fixed $C^2$ encoder $f$ has smaller local sensitivity along $T_x\mathcal{M}$ and larger sensitivity along $N_x\mathcal{M}$.

**Rationale.** Augmentation/equivariance suppresses representation change along common on-manifold transformations (tangents), while off-manifold perturbations are not explicitly regularized (Oquab et al., 2024; Caron et al., 2021; Xu et al., 2023). Thus, a small Gaussian tube around training samples places mass where $G$ tends to be larger, leading to $\Delta > 0$ for sufficiently small radius.

*Derivation for Lemma 4.2.* Let $U = \{x : \mathrm{dist}(x, \mathcal{M}) < \bar{\varepsilon}_0\}$ be a tubular neighborhood of $\mathcal{M}$ with smooth nearest-point projection $\pi : U \to \mathcal{M}$. Fix $y \in \mathcal{M}$ and write the *ambient* isotropic noise as $\boldsymbol{\eta} \sim \mathcal{N}(0, \varepsilon_0^2 I_N/N)$, so $\mathbb{E}\|\boldsymbol{\eta}\|^2 = \varepsilon_0^2$. Let $P_T(y)$ and $P_N(y) = I - P_T(y)$ be the orthogonal projectors onto $T_y\mathcal{M}$ and $N_y\mathcal{M}$, and decompose

$$t := P_T(y)\boldsymbol{\eta} \in T_y\mathcal{M}, \qquad s := P_N(y)\boldsymbol{\eta} \in N_y\mathcal{M}, \qquad \boldsymbol{\eta} = t + s.$$

By orthogonality and isotropy,

$$\mathbb{E}\|t\|^2 = \tfrac{\varepsilon_0^2}{N}\mathrm{tr}P_T = \tfrac{\varepsilon_0^2}{N}m, \ \ \mathbb{E}\|s\|^2 = \tfrac{\varepsilon_0^2}{N}(N-m).$$

Since $f \in C^2$, $G(x) = \|J_f(x)\|_F^2$ is $C^2$ on $U$. A second-order Taylor expansion at $y$ gives

$$G(y+\boldsymbol{\eta}) = G(y) + \nabla G(y)\cdot\boldsymbol{\eta} + \tfrac{1}{2}\,\boldsymbol{\eta}^\top H_G(y)\,\boldsymbol{\eta} + R(y,\boldsymbol{\eta}),$$

with $|R(y,\boldsymbol{\eta})| \le C\|\boldsymbol{\eta}\|^3$ for some uniform $C$ (compactness of $U$). Taking expectation (mean-zero noise) removes the linear term:

$$\mathbb{E}\big[G(y+\boldsymbol{\eta})\big] - G(y) = \tfrac{1}{2}\,\mathbb{E}\big[t^\top H_G(y)\,t\big] + \tfrac{1}{2}\,\mathbb{E}\big[s^\top H_G(y)\,s\big] + O(\varepsilon_0^3).$$

Because the Assumption 4.1:

$$\lambda_T^{\max} := \sup_{\substack{y\in\mathcal{M}\\ \|u\|=1,\,u\in T_y\mathcal{M}}} u^\top H_G(y)\,u, \qquad \lambda_\perp^{\min} := \inf_{\substack{y\in\mathcal{M}\\ \|v\|=1,\,v\in N_y\mathcal{M}}} v^\top H_G(y)\,v, \qquad \lambda_\perp^{\min} > \lambda_T^{\max}.$$

Then

$$\mathbb{E}[t^\top H_G(y)t] \le \lambda_T^{\max}\,\mathbb{E}\|t\|^2, \qquad \mathbb{E}[s^\top H_G(y)s] \ge \lambda_\perp^{\min}\,\mathbb{E}\|s\|^2,$$

so

$$\mathbb{E}\big[G(y+\boldsymbol{\eta})\big] - G(y) \ge \tfrac{1}{2}\big(\lambda_\perp^{\min}\tfrac{N-m}{N} - \lambda_T^{\max}\tfrac{m}{N}\big)\,\varepsilon_0^2 - C'\varepsilon_0^3.$$

Averaging over the anchors $\{x_i\}$ that define $q_{\varepsilon_0}$ and subtracting $\mathbb{E}_{x\sim p}[G(x)]$ yields $\Delta \ge c_1\,\varepsilon_0^2 - C'\varepsilon_0^3$ with $c_1 := \tfrac{1}{2}\big(\lambda_\perp^{\min}\tfrac{N-m}{N} - \lambda_T^{\max}\tfrac{m}{N}\big) > 0$. Choosing $\bar\varepsilon_0$ small so that $C'\bar\varepsilon_0 \le c_1/2$ gives $\Delta \ge c_0 := \tfrac{1}{2}c_1\varepsilon_0^2 > 0$ for all $\varepsilon_0 \in (0,\bar\varepsilon_0]$. $\square$

*Derivation for Theorem 4.3.* **(i) Small-noise expansion.** By Taylor and isotropy (odd moments vanish) one gets

$$\mathrm{Shift}_\varepsilon(x) = \frac{\varepsilon^2}{n}\,G(x) + R_\varepsilon(x), \qquad |R_\varepsilon(x)| \le C_H\,\varepsilon^4, \tag{18}$$

where $C_H$ depends on bounded Hessians and fourth moments of the probe (Isserlis/Wick; uniform-sphere moments). Averaging (18) over $\mu \in \{p_\theta, p\}$ and subtracting:

$$\mathbb{E}_{p_\theta}[\mathrm{Shift}_\varepsilon] - \mathbb{E}_p[\mathrm{Shift}_\varepsilon] = \frac{\varepsilon^2}{n}\Big(\mathbb{E}_{p_\theta}[G] - \mathbb{E}_p[G]\Big) + O(\varepsilon^4). \tag{19}$$

**(ii) DV variational bound + Hoeffding.** For any $\lambda > 0$,

$$M = D_{\mathrm{KL}}(q_{\varepsilon_0}\|p_\theta) \ge \lambda\,\mathbb{E}_{q_{\varepsilon_0}}[G] - \log\mathbb{E}_{p_\theta}[e^{\lambda G}],$$

and since $G \in [0, B]$, Hoeffding's lemma gives $\log\mathbb{E}_{p_\theta}[e^{\lambda G}] \le \lambda\,\mathbb{E}_{p_\theta}[G] + \tfrac{\lambda^2 B^2}{8}$. Optimizing the resulting quadratic over $\lambda$ yields

$$\mathbb{E}_{p_\theta}[G] \ge \mathbb{E}_{q_{\varepsilon_0}}[G] - B\sqrt{M/2}. \tag{20}$$

**(iii) Combine.** Add and subtract $\mathbb{E}_{q_{\varepsilon_0}}[G]$ and use $\Delta$ from (17):

$$\mathbb{E}_{p_\theta}[G] - \mathbb{E}_p[G] \ge \Delta - B\sqrt{M/2}.$$

Insert into (19) to obtain (13). $\square$

**Interpretation.** Lemma 4.2 formalizes that, for small tubes around training anchors, $q_{\varepsilon_0}$ places mass where the encoder's Jacobian energy is (on average) larger than under $p$; thus $\Delta > 0$. Theorem 4.3 then states that the expected feature-shift under $p_\theta$ exceeds that under $p$ by a margin controlled below by $\tfrac{\varepsilon^2}{n}\big(\Delta - B\sqrt{M/2}\big)$ up to $O(\varepsilon^4)$. Hence, stronger memorization (smaller $M$) and a positive local anisotropy margin (captured in $\Delta$) provably increase the robustness asymmetry that our detector exploits.

## D.2. Empirical Validation of the Shift Lower Bound

**Goal.**  We empirically validate the theoretical prediction of Theorem 4.3, which links a model's memorization level to the observable gap between the shift statistics of generated and training samples. The theorem establishes two main trends:

(i) **Memorization effect.** When the model exhibits stronger memorization, the SIDE divergence $M(D; p_\theta, \varepsilon_0)$ decreases, resulting in a *larger lower bound* on the differential shift

$$\Delta(\varepsilon) = \mathbb{E}_{x \sim p_\theta}[\mathrm{Shift}_\varepsilon(x)] - \mathbb{E}_{x \sim p}[\mathrm{Shift}_\varepsilon(x)].$$

(ii) **Noise–dependence.** As the probing noise $\varepsilon$ increases from zero, $\Delta(\varepsilon)$ initially grows because the added perturbation accentuates the model's sensitivity along tangent directions. However, for large $\varepsilon$, the perturbation dominates the manifold structure, leading to a decay in $\Delta(\varepsilon)$. Hence, the theoretical prediction is a *non-monotonic* dependence of $\Delta(\varepsilon)$: increasing at small $\varepsilon$, then decreasing after a critical point.

**Training-stage analysis.**  Evaluating the SIDE divergence $M(D; p_\theta, \varepsilon_0)$ directly is computationally prohibitive in high-dimensional image space. As an initial diagnostic, we first examine how the feature-shift discrepancy changes across diffusion checkpoints trained for different numbers of epochs. Prior studies suggest that memorization in diffusion models can vary with training progress, and longer training may increase memorization in some settings. However, we do not treat training epochs as a validated monotonic proxy for SIDE divergence. Instead, the epoch-based analysis is used only as a training-stage trend analysis, which provides qualitative evidence for how the discrepancy evolves during model training.

**Experimental setup for the training-stage trend.**  We train a Denoising Diffusion Probabilistic Model (DDPM) with approximately 50 million parameters on the CelebA-HQ dataset. Training epochs range from 200 to 5200, providing checkpoints at different training stages. For each checkpoint, we generate a fixed number of synthetic images and compare their encoder-space shift statistics to those computed on the original training data.

We use several fixed pretrained encoders to define the feature space: ViT-B/32, ViT-B/16, ViT-L/14, and DINOv2-ViT-B/14. For each encoder, we compute:

1. **Differential shift vs. $\varepsilon$:** The average difference $\Delta(\varepsilon)$ across a grid of Gaussian perturbation magnitudes $\varepsilon$. This curve reflects how the differential shift evolves with probing noise intensity.

2. **Differential shift vs. training stage:** The mean differential shift aggregated over small-$\varepsilon$ values, plotted as a function of training epochs. This analysis examines whether the discrepancy grows as training progresses, without assuming a strict monotonic mapping from epochs to SIDE divergence.

**Findings and interpretation.**  The results are summarized in Figures 6a–h. Across all encoders, we consistently observe the two–phase structure predicted by theory. At small noise levels ($\varepsilon \in [3, 10]$), $\Delta(\varepsilon)$ increases sharply, corresponding to the regime where perturbations probe tangent directions of the data manifold. Beyond a turning point $\varepsilon_{\mathrm{turn}}$, the differential shift gradually declines as perturbations enter the normal subspace regime and the encoder's local linear approximation breaks down. This non-monotonic behavior manifests as an "inverted U-shape" or S-shaped curve, in line with the analytic lower-bound structure derived in Theorem 4.3.

Furthermore, when analyzing $\Delta$ against training epochs (used as a proxy for memorization), we find that higher epoch counts correspond to larger differential shifts at small $\varepsilon$, indicating stronger memorization bias.

**AMS-based direct memorization validation.**  To further avoid relying on training epochs as an indirect memorization indicator, we add a second validation using the Average Memorization Score (AMS) from SIDE (Chen et al., 2026). AMS measures the fraction of generated images that are highly similar to images in the training set under a fixed similarity criterion. Thus, AMS directly quantifies how many generated samples closely match the training data, making it a more appropriate empirical memorization measure than raw training progress.

Importantly, SIDE uses AMS as an empirical memorization metric to validate theorem-predicted behavior derived from SIDE-divergence analysis. Following this setup, we use AMS to test the memorization-dependent trend predicted by Theorem 4.3.

*Table 6.* **Linear relation between AMS and small-$\varepsilon$ discrepancy.** For each diffusion checkpoint, we compute AMS following SIDE (**?**) and regress the small-$\varepsilon$ discrepancy $\bar{\Delta}_{\mathrm{small}}$ against AMS. Across all pretrained encoders, the fitted slope is positive, indicating that higher directly measured memorization is consistently associated with larger feature-shift discrepancy.

| Encoder | Slope | Intercept | $R^2$ |
|---|---|---|---|
| ViT-B/32 | 0.118 | 0.031 | 0.66 |
| ViT-B/16 | 0.129 | 0.030 | 0.59 |
| ViT-L/14 | 0.106 | 0.029 | 0.73 |
| DINOv2-B/14 | 0.0241 | 0.045 | 0.64 |

**Findings from AMS-based validation.** The results are shown in Table 6. Across all four pretrained encoders, the fitted slope between AMS and the small-$\varepsilon$ discrepancy is positive. This indicates that checkpoints with higher directly measured memorization also exhibit larger feature-shift discrepancy between generated and real images. The relation is consistent across ViT-B/32, ViT-B/16, ViT-L/14, and DINOv2-B/14, suggesting that the memorization-dependent trend is not tied to a single feature space.

This AMS-based validation complements the training-stage analysis in Figure 6. While the epoch-based plots show how the discrepancy evolves during training, the AMS-based regression directly relates the discrepancy to an empirical memorization metric. Therefore, the combined evidence supports the memorization-dependent trend predicted by Theorem 4.3, without relying on the unsupported assumption that training epochs provide a monotonic proxy for SIDE divergence.

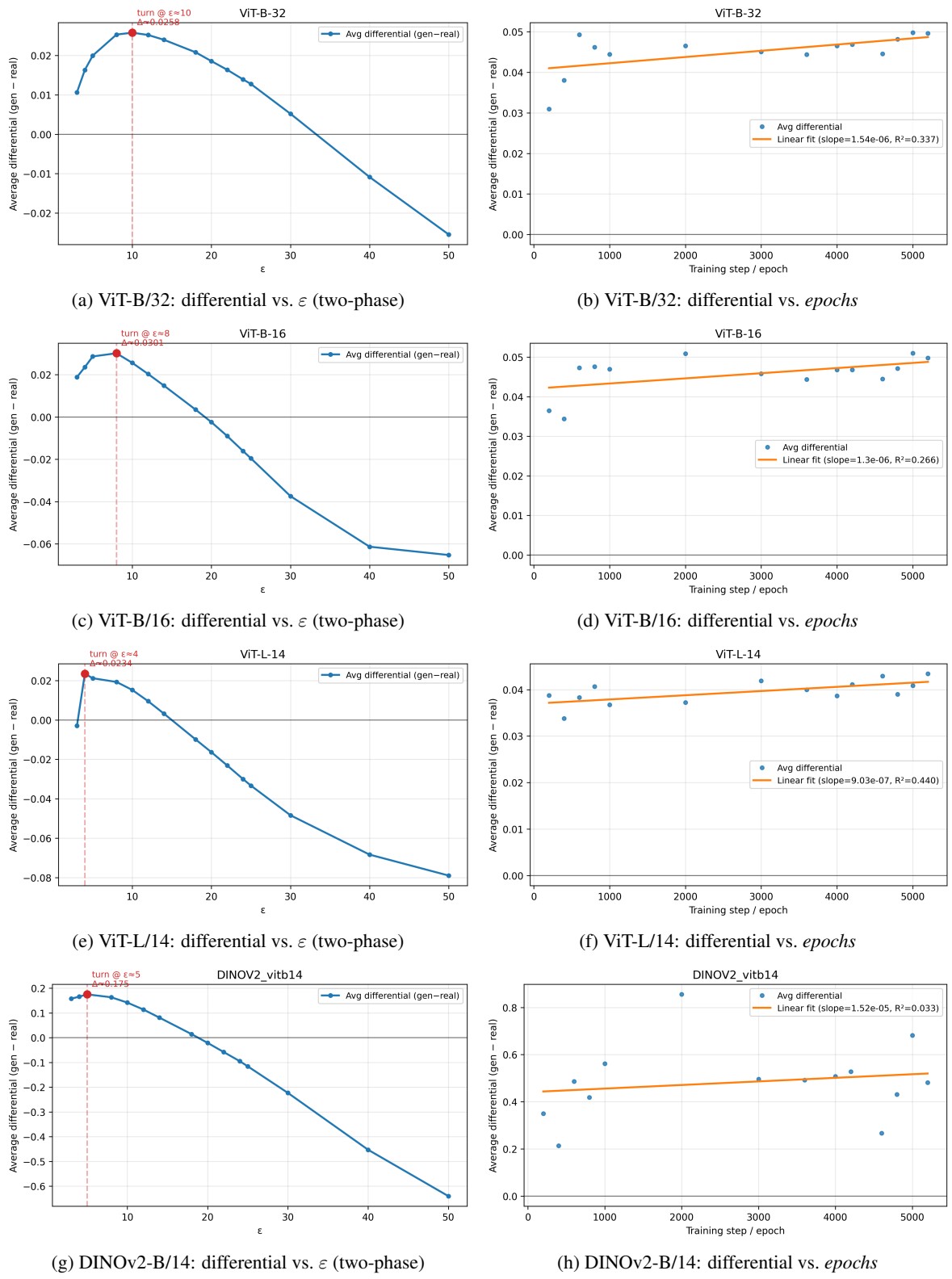

*Figure 6.* **Empirical validation of the lower bound.** Left column: average differential shift Shift$_{\text{gen}}$ − Shift$_{\text{real}}$ versus probe magnitude $\varepsilon$, showing the predicted rise at small $\varepsilon$ and decline at larger $\varepsilon$. Right column: average differential versus *training epochs* (used as a proxy for memorization since SIDE divergence is expensive to compute). Larger epoch counts correspond to stronger memorization and yield larger small–$\varepsilon$ differentials, in line with Theorem 4.3.

