# OpenReview forum: "RA-Det: Towards Universal Detection of AI-Generated Images via Robustness Asymmetry"
_ICML.cc/2026/Conference — ICML 2026 regular_

### Official Review · Reviewer_Zz32 · 2026-03-12

**Soundness:** 3
**Presentation:** 4
**Significance:** 3
**Originality:** 3
**Overall Recommendation:** 4
**Confidence:** 4

**Summary:**

To address the detection bottlenecks caused by high-quality generative images, this paper proposes a novel paradigm shift from "appearance feature analysis" to "behavioral probing." The authors point out that many existing detection methods rely primarily on visual appearance cues, such as forensic artifacts or statistical distribution discrepancies. However, as the capabilities of generative models advance, these cues are becoming increasingly inconspicuous, thereby undermining the stability and generalization of detection methods across new models. To tackle this issue, the paper explores a new perspective by analyzing the behavioral response of images under small-scale perturbations. The authors identify a phenomenon termed "Robustness Asymmetry": when small, structured perturbations are applied to natural images, their semantic representations typically remain stable, whereas the feature representations of generated images exhibit significantly larger drift. The paper further provides a theoretical discussion of this phenomenon, linking it to the potential memorization tendencies of generative models. Consequently, the authors propose RA-Det, a detection framework that quantifies the degree of feature drift induced by active, structured perturbations, using this discrepancy as a discriminative signal to distinguish between real and generated images. Experimental results demonstrate that the proposed method achieves robust generalization across a variety of mainstream generators, with a 12.92% improvement in mean accuracy compared to existing methods.

**Compliance With Llm Reviewing Policy:**

Affirmed.

**Key Questions For Authors:**

1.	The "robustness asymmetry" observed in the paper is primarily measured within the feature spaces of foundation models with strong semantic priors, such as CLIP. If the feature space is replaced with a model that lacks such strong semantic priors, would this "robustness asymmetry" remain significant?
2.	Compared to undifferentiated simple perturbations, how much specific performance gain does the condition-based targeted probing of the DRP module provide? Could a set of comparative experiments be provided to demonstrate the necessity and superiority of "targeted probing" in distinguishing between real and fake images?

**Limitations:**

The authors should further explore the stability of the framework against adaptive adversarial attacks. If an attacker gains knowledge of the DRP (Differential Robustness Probing) mechanism, it is entirely possible to perform adversarial fine-tuning on the generative models, enabling the generated images to acquire a similar "semantic resilience." Would such a customized attack targeting behavioral features lead to the failure of the "robustness asymmetry" as a discriminative signal?

**Strengths And Weaknesses:**

Strength
The shift of detection logic from image appearance features to "behavioral features" represents a highly novel paradigm shift, which effectively avoids the dilemma where generative models increasingly approach reality at the pixel level. Furthermore, the paper provides not only empirical results but also derives a theoretical lower bound for "robustness asymmetry," tracing it back to the memorization essence of generative models, which enhances the methodological rigor. Compared to adding random noise, the DRP module utilizes a conditional network to generate "structured perturbations" to induce feature drift; this design ensures that the discriminative power of the detection signal is far superior to that of passive observation. The authors also conducted tests against JPEG compression and Gaussian blur, demonstrating that this method possesses superior survival capabilities in common post-processing scenarios compared to traditional methods.

Weakness
Measuring "robustness asymmetry" is highly dependent on the embedding space of specific foundation models (e.g., CLIP). Since foundation models are exposed to vast amounts of real-world data during the pre-training phase, their feature spaces may naturally be smoother toward real images. It remains to be further discussed whether this asymmetry is an inherent flaw of generative images or an artifact resulting from the foundation model's overfitting to natural data. The paper attributes this phenomenon to "memorization tendencies," yet the experiments lack a controlled variable analysis regarding the degree of model memorization.

---

> ### Author Rebuttal · Authors · 2026-03-31
>
> We sincerely thank Reviewer Zz32 for the constructive feedback and insightful comments. We appreciate the recognition of our paradigm shift, theoretical grounding, and robustness analysis. We address the concerns below.
>
> ---
>
> > Q1. Is robustness asymmetry inherent, or dependent on foundation models with strong semantic priors?
>
> **R1.** Thank you for this insightful question. We clarify that robustness asymmetry arises from a synergistic interaction between the foundation model and the inherent flaws of generative images.
>
> As stated in Assumption 4.1, foundation models naturally learn a smoother feature space along the real-image manifold due to large-scale pre-training, making them less sensitive to perturbations in these regions. On the other hand, generative models imperfectly approximate this manifold and exhibit off-manifold deviations and memorization behavior. Importantly, this encoder bias is not a confounding artifact but a necessary lens to expose generative imperfections. As proven in Theorem 4.3, the magnitude of robustness asymmetry is lower-bounded by the generator’s memorization (SIDE divergence), indicating that the phenomenon is fundamentally driven by the generator rather than the encoder. We further provide controlled empirical validation (Appendix B.2, Fig. 6): by fixing the encoder and varying generator training epochs (200→5200) as a proxy for memorization, we observe that the robustness gap consistently increases with memorization. This confirms that the asymmetry is actively driven by synthetic images, not solely dictated by the feature space. Finally, we observe consistent behavior across different pre-training paradigms (e.g., CLIP and DINOv3), suggesting that robustness asymmetry is a universal interaction between natural-image-biased manifolds and generative imperfections. We will clarify this interplay more explicitly in the revision.
>
> ---
> >Q2. Compared to undifferentiated simple perturbations, how much benefit does condition-based targeted probing provide? Could comparative experiments demonstrate the necessity and superiority of targeted probing?
>
> **R2.** Thank you for this important question. Simple perturbations were our starting point. As shown in Table 9, Gaussian noise can indeed reveal robustness asymmetry, but the gap is highly unstable across noise levels and generators, indicating that passive perturbations do not provide a reliable detection signal.(Full statistics in Table 2 of https://anonymous.4open.science/r/RA-Det-Rebuttal-Appendix-75F7/)
>
> **Table 9. Robustness asymmetry gap under Gaussian perturbations.**
>
> | Generator  | ($\sigma=0.03$) | ($\sigma=0.05$) | ($\sigma=0.10$) |
> | ---------- | --------------: | --------------: | --------------: |
> | ProGAN     |          0.1052 |          0.1369 |          0.1651 |
> | GauGAN     |          0.0990 |          0.1236 |          0.1422 |
> | ADM        |          0.0488 |          0.0636 |          0.0798 |
> | Midjourney |          0.0145 |          0.0158 |          0.0097 |
>
> In contrast, DRP learns condition-based perturbations to actively amplify the real/fake discrepancy. As shown in Table 10, DRP consistently outperforms standard perturbations, confirming that the gain comes from targeted probing, rather than simply adding noise.
>
> **Table 10. Comparison of different perturbations.**
>
> | Method          | ACC (%) | AP (%) |
> | --------------- | ------: | -----: |
> | DRP             |   93.47 |  97.02 |
> | Gaussian Noise  |   86.11 |  91.71 |
> | Gaussian Blur   |   87.07 |  91.42 |
> | Resize Residual |   86.59 |  90.83 |
>
> ---
> > Q3. Robustness against adaptive attacks targeting behavioral features.
>
> **R3.** Thank you for raising this important concern. We agree that robustness against adaptive attacks is an important direction. We emphasize that robustness asymmetry stems from structural limitations of generative models, rather than superficial artifacts. As shown in Theorem 4.3, this asymmetry is fundamentally linked to memorization effects and imperfect modeling of the natural image manifold. To eliminate this asymmetry, an attacker would need to ensure that generated images exhibit the same stability as real images under structured perturbations. This would require addressing the manifold learning gap between generated and real data, which goes beyond standard adversarial fine-tuning. Therefore, while adaptive attacks may partially reduce the signal, fully removing robustness asymmetry would require fundamentally improving generative models themselves. We will include a more detailed discussion of this limitation in the revision.
>
> ---
>
> We thank the reviewer again for the valuable feedback. We will incorporate these clarifications and analyses in the revised manuscript.

---

> > ### Author Rebuttal · Reviewer_Zz32 · 2026-04-03
> >
> > I am generally satisfied with the author's rebuttal, but I still hope to see some preliminary experimental results regarding the issue of adaptive attacks.

---

> > > ### Author Response · Authors · 2026-04-04
> > >
> > > Thank you for the follow-up and for encouraging us to provide preliminary experimental evidence on adaptive attacks.
> > >
> > > We further add a preliminary adaptive attack study. Specifically, we construct customized attacks against RA-Det that explicitly target the robustness asymmetry signal induced by DRP. We evaluate both FGSM and PGD under the perturbation setting ($\epsilon$=4/255; for PGD, $\alpha$=1/255 and 10 steps).
> > >
> > > **Table 11. Robustness of RA-Det under adaptive attacks.**
> > >
> > > | Method | Setting | Mean ACC | Mean AP |
> > > | ------ | ------- | -------- | ------- |
> > > | RA-Det | clean   | 93.47    | 97.00   |
> > > | RA-Det | FGSM    | 85.73    | 83.28   |
> > > | RA-Det | PGD     | 69.38    | 70.30   |
> > >
> > > As shown, RA-Det remains robust under both attacks. Even under PGD, RA-Det still retains 69.38 ACC / 70.30 AP. This suggests that the robustness asymmetry signal cannot be easily removed by standard adversarial optimization alone.
> > >
> > > We believe this mainly comes from two reasons. First, robustness asymmetry is rooted in structural deficiencies of generated images, rather than superficial artifacts, so adversarial perturbations may weaken but cannot fundamentally eliminate the signal. Second, RA-Det uses complementary multi-branch information rather than a single fragile cue, which further improves robustness under attack.
> > >
> > > We thank the reviewer again for raising this important perspective. We will clarify this limitation and include it in the revised paper.

---

### Official Review · Reviewer_GbMZ · 2026-03-12

**Soundness:** 3
**Presentation:** 3
**Significance:** 3
**Originality:** 3
**Overall Recommendation:** 4
**Confidence:** 3

**Summary:**

The paper proposes RA-Det, a new framework to detect generated images, by leveraging the insight that generated images experience a larger shift in a feature embedding space when perturbed compared to natural images. The framework introduces a new approach on analyzing how images behave under perturbations, rather than analyzing solely the image quality.

**Compliance With Llm Reviewing Policy:**

Affirmed.

**Final Justification:**

The proposed RA-Det method detects generated data, by analyzing *how* the data behaves rather than what the data looks like. The rebuttal addressed my main concerns and showed that the method also performs well on newer generative models, outperforming existing baselines.

There are clear merits to the paper that support accepting this work.

**Key Questions For Authors:**

1. Could the Differential Robustness Probing component be replaced by standard image augmentations such as Gaussian Noise with fixed mean and variance, Gaussian Blur or other augmentations?

2. How well does the method perform on newer generative models such as StableDiffusion v3.5 and FLUX.1 and on autoregressive models, such as Infinity and VAR?

3. Which real data was used for the evaluation of the method?

**Limitations:**

yes

**Strengths And Weaknesses:**

### Strengths:
- The idea of leveraging the behavior of the image is well principled and supported by a proof on the lower bound for the robustness asymmetry.
- The paper provides extensive experimental results with an insightful ablation study.
- The paper is well written and easy to follow.

### Weaknesses:
- The experimental evaluation focuses strongly on older models and GAN.
	- It should be extended to newer generative models such as Stable Diffusion v3.5 and FLUX.1.
	- The method should also be evaluated on the autoregressive generation domain with models such as Infinity [1] or VAR [2].
- The experimental setup is unclear in regards to which real data was used during training and during evaluation.
- To gain additional insights into the reliability of the framework, the paper should also report the TPR@1%FPR for the results obtained in Table 1.


**References**

[1] Jian Han et al. “Infinity: Scaling Bitwise AutoRegressive Modeling for High-Resolution Image Synthesis”. CVPR. 2025.

[2] Keyu Tian et al. “Visual Autoregressive Modeling: Scalable Image Generation via Next-Scale Prediction”. NeurIPS. 2024.

---

> ### Author Rebuttal · Authors · 2026-03-31
>
> We sincerely thank Reviewer GbMZ for the constructive feedback and insightful comments. We address the concerns below.
>
> ------
>
> > Q1. How does the method perform on newer generative models (e.g., SD3.5, FLUX.1, Infinity, VAR)?
>
> **R1.** We agree that evaluating on newer generators is important for assessing the practical relevance of RA-Det. Following suggestion, we extend our evaluation to recent diffusion and autoregressive models, including SD3.5, FLUX.1, Infinity, and VAR.
>
> **Table 6. Evaluation on recent generative models(ACC/AP).**
>
> | Model  |      SDv3.5 |      FLUX.1 |    Infinity |         VAR |
> | ------ | ----------: | ----------: | ----------: | ----------: |
> | UnivFD | 60.10/71.91 | 38.99/38.10 | 66.13/82.40 | 68.62/85.05 |
> | NPR    | 43.60/43.20 | 53.63/65.99 | 59.52/68.40 | 39.47/40.71 |
> | RA-Det | 75.61/81.65 | 80.58/88.40 | 80.09/88.32 | 84.16/91.63 |
>
> Despite being trained only on ProGAN, RA-Det still maintains reasonable performance on these unseen and architecturally diverse generators. We do observe a performance drop compared with earlier generators, which is expected because newer models often produce images that lie closer to the natural image manifold and therefore reduce the separability of both artifact-based and behavior-based cues. Nevertheless, RA-Det continues to generalize without relying on generator-specific fingerprints, which supports our central claim that robustness asymmetry provides a model-agnostic detection signal.
>
> In the revision, we will include these additional comparisons and also add the corresponding citations for the newer generators, including Infinity [1] and VAR [2].
>
> ------
>
> > Q2. Could DRP be replaced by standard augmentations (e.g., Gaussian noise, blur)?
>
> **R2.** Thank you for this important question. We compare DRP with several standard perturbations, including Gaussian noise, Gaussian blur, and resize residual.
>
> **Table 7. Comparison with standard perturbation.**
>
> | Method          | ACC (%) | AP (%) |
> | --------------- | ------- | ------ |
> | DRP             | 93.47   | 97.02  |
> | Gaussian Noise  | 86.11   | 91.71  |
> | Gaussian Blur   | 87.07   | 91.42  |
> | Resize Residual | 86.59   | 90.83  |
>
> The gap is clear and consistent: DRP substantially outperforms these generic perturbations. This suggests that the gain does not come from simply adding perturbations, but from **targeted probing** that more effectively exposes robustness asymmetry while preserving semantic content. By contrast, standard augmentations are fixed and task-agnostic, making the resulting signal less stable and less discriminative.
>
> We will clarify this point in the revision. More detailed analysis, including the comparison between passive Gaussian perturbations and condition-based probing, is provided in our response to **Reviewer Zz32, Q2**.
>
> ------
>
> > Q3. Report TPR@1%FPR for Table 1.
>
> **R3.** Thank you for the suggestion. We report TPR@1%FPR below, with full per-generator results available in the anonymous appendix (see *Table 1 (c) TPR at 1% FPR (↑)*). Overall, RA-Det consistently outperforms prior methods on both original and additional generators under strict low-FPR constraints. The relatively low TPR values indicate that reliable detection in this regime remains challenging and requires further study.
>
> **Table 8. TPR@1%FPR comparison.**
>
> | Setting                         | UniFD | NPR  | Ours  |
> | ------------------------------- | ----- | ---- | ----- |
> | Original Generators TPR@1%FPR   | 63.42 | 8.39 | 78.65 |
> | Additional Generators TPR@1%FPR | 7.25  | 4.02 | 31.81 |
>
> Full results: https://anonymous.4open.science/r/RA-Det-Rebuttal-Appendix-75F7/README.md
>
> ------
>
> > Q4. Which real data is used for training and evaluation?
>
> **R4.** Thank you for this important question. Following the widely adopted protocol in CNNSpot [3] and PatchCraft [4], our training and evaluation sets use non-overlapping image splits.
>
> Specifically, the training set consists of **360K real images from LSUN and 360K fake images from ProGAN**, following prior works. For evaluation, the real images are drawn from the benchmark test sets associated with each generator and domain, including datasets such as ImageNet, LSUN, CelebA, COCO, and FFHQ. This protocol ensures fair comparison and avoids data leakage between training and testing.
>
> We will revise the paper to state this protocol more explicitly.
>
> Overall, we will revise the paper accordingly to improve the completeness and clarity of the empirical evaluation and experimental protocol.
>
> **References**
>
> [1] Jian Han et al. *Infinity: Scaling Bitwise AutoRegressive Modeling for High-Resolution Image Synthesis*. CVPR, 2025.
> [2] Keyu Tian et al. *Visual Autoregressive Modeling: Scalable Image Generation via Next-Scale Prediction*. NeurIPS, 2024.
> [3] Wang et al. *CNN-generated images are surprisingly easy to spot... for now*. CVPR, 2020.
> [4] Zhong et al. *PatchCraft: Exploring Texture Patch for Efficient AI-generated Image Detection*. arXiv, 2023.

---

> > ### Author Rebuttal · Reviewer_GbMZ · 2026-04-03
> >
> > I would like to thank the authors for their extensive rebuttal.
> >
> > RA-Det outperforms the baselines on newer generative models as well as on the autoregressive image generation domain. The ablation shows that the DRP is a necessary component that enables strong performance across datasets. The used protocol enables a fair comparison and avoids data leakage, which supports the reported results.
> >
> > Overall I have decided to increase my score.

---

> > > ### Author Response · Authors · 2026-04-03
> > >
> > > Thank you very much for your thoughtful feedback and for your positive update. We would greatly appreciate it if you could kindly reflect this score change in the original review in the system as well.

---

### Official Review · Reviewer_GSp9 · 2026-03-14

**Soundness:** 4
**Presentation:** 4
**Significance:** 3
**Originality:** 3
**Overall Recommendation:** 4
**Confidence:** 4

**Summary:**

The paper proposes RA-Det, a framework for detecting AI-generated images. It operates on the principle of "robustness asymmetry," observing that synthetic images exhibit larger feature-space displacement than real images when subjected to small, controlled perturbations. The method employs a learned conditional UNet (Differential Robustness Probing) to generate bounded pixel-space perturbations. Both original and perturbed images are processed through a frozen DINOv3 ViT-L/16 encoder to extract embeddings. A multi-branch detector then aggregates semantic features, discrepancy metrics (distance and difference between embeddings), and low-level pixel residuals to classify the image. The authors provide a theoretical lower bound linking this feature shift to generative model memorization using the SIDE divergence. Evaluation is conducted on the PatchCraft benchmark, training solely on ProGAN and testing on 15 unseen models.

**Compliance With Llm Reviewing Policy:**

Affirmed.

**Key Questions For Authors:**

1. Can the authors provide detection results on a test set containing modern generators (e.g., SDXL, Midjourney v6, FLUX)?

**Limitations:**

The authors have not adequately discussed the limitations of their work.

**Strengths And Weaknesses:**

**Strengths:**

* **Originality:** The shift from appearance-based artifact detection to behavior-based probing using learned perturbations offers an interesting angle for deepfake detection.
* **Presentation:** The motivation is clearly articulated. The multi-branch architecture logically follows the premise of capturing both high-level representation drift and low-level pixel residuals.
* **Significance:** The method achieves a high mean accuracy (93.47%) on the chosen benchmark.

**Weaknesses:**

* **Soundness (Backbone Confound):** The performance gains are heavily conflated with the use of the DINOv3 ViT-L/16 backbone, a highly capable foundation model. The ablation study indicates that the semantic branch alone achieves 90.34% accuracy. Baselines are not evaluated with this same feature extractor, making the claimed 12.92% improvement misleading.
* **Significance (Evaluation Age):** The evaluation benchmark lacks modern state-of-the-art generative models (e.g., SDXL, Stable Diffusion 3, Midjourney v6, FLUX). Claims of "universal detection" are unsupported when tested primarily on older GANs and early diffusion models.
* **Soundness (Computational Cost):** The method requires a UNet forward pass, two passes through a ViT-L/16, and three detection heads per image. No inference time, FLOPs, or memory metrics are provided. This omission is critical given the practical constraints of scale in deepfake detection.
* **Soundness (Theory vs. Practice):** The theoretical lower bound relies heavily on Assumption 4.1 (anisotropy in encoder sensitivity), which borders on tautological for the claimed result. Furthermore, the empirical validation of the bound substitutes training epochs as a proxy for SIDE divergence, which weakens the empirical backing of the theoretical claims.
* **Originality / Soundness (Missing Baselines & Ablations):** The evaluation omits recent relevant works like DRCT, which also uses perturbation and reconstruction techniques. Additionally, there is no ablation comparing the learned DRP perturbation to simple random noise (e.g., Gaussian).

---

> ### Author Rebuttal · Authors · 2026-03-31
>
> We sincerely thank Reviewer GSp9 for the constructive feedback and insightful comments. We appreciate the recognition of our motivation, design, and empirical performance. We address the concerns below.
>
> ---
>
> > Q1. Backbone confound.
>
> We thank the reviewer for the comment. The reported 90.34% accuracy in Table 2 corresponds to “w/o Semantic Branch”, not the semantic branch alone, indicating that performance is not dominated by the backbone. Moreover, replacing DINOv3 with CLIP still yields 91.85% ACC, demonstrating robustness across backbones. Notably, UnivFD also uses CLIP ViT-L/14, and under the same backbone, our method still outperforms it. This suggests that the performance gain mainly comes from the proposed robustness-asymmetry modeling rather than the backbone choice.
>
> ---
>
> > Q2. Evaluation lacks modern generative models (e.g., SDXL, Midjourney v6, FLUX).
>
> We agree with the reviewer and extend our evaluation to more recent generators.
>
> **Table 4: Evaluation results on modern generative models.**
>
> | Model           |        SDXL |      SDv3.5 |      FLUX.1 | Midjourney v6 |     Average |
> | --------------- | ----------: | ----------: | ----------: | ------------: | ----------: |
> | UnivFD          | 55.34/57.42 | 60.10/71.91 | 38.99/38.10 |   25.12/33.59 | 44.89/50.26 |
> | NPR             | 48.23/51.02 | 43.60/43.20 | 53.63/65.99 |   39.98/42.82 | 46.36/50.76 |
> | DRCT            | 82.87/86.67 | 80.45/81.67 | 70.52/72.51 |   50.03/45.73 | 70.97/71.65 |
> | RA-Det          | 96.29/91.34 | 75.61/81.65 | 80.58/88.40 |   74.97/70.93 | 81.86/83.08 |
> | DRCT(SD-v1.4)   | 94.01/94.01 | 91.92/91.92 | 77.60/77.60 |   80.04/80.04 | 85.89/85.89 |
> | RA-Det(SD-v1.4) | 94.42/98.95 | 91.85/96.14 | 90.00/96.44 |   81.23/89.96 | 89.38/95.37 |
>
> Despite being trained only on ProGAN, RA-Det generalizes well to modern generative models. When additionally trained with SD-v1.4, the performance is further improved. Although performance drops on more advanced models, likely due to their closer alignment with the natural image manifold, RA-Det still maintains robust effectiveness without relying on generator-specific artifacts.
>
> ---
>
> > Q3. Missing efficiency analysis (inference time, FLOPs, memory).
>
> Thank you for pointing out this important aspect. We have supplemented the experiments and analysis; detailed efficiency comparisons can be found in Table 1 of our reply to **Reviewer WPGe**.
>
>
> ---
>
> > Q4. Theoretical assumptions and empirical validation are not well aligned.
>
> We respectfully disagree that Assumption 4.1 makes the result tautological. Assumption 4.1 specifies a local geometric property of the pretrained encoder and is used to establish a positive anisotropy margin ($\Delta$). It does not directly yield the lower bound in Theorem 4.3, nor does it by itself connect robustness asymmetry to memorization. The key contribution of Theorem 4.3 is the nontrivial derivation that links the robustness gap to the SIDE memorization term ($M(D; p_\theta, \varepsilon_0)$). In other words, the memorization-dependent lower bound is not assumed; it is obtained through intermediate mathematical development built on Assumption 4.1. We agree that training epochs are not a direct measurement of SIDE divergence, and we will revise the paper to state this more explicitly. Our use of epochs is intended only as a practical proxy for memorization dynamics. This choice is supported by prior works[1,2] showing that memorization is highly correlated with training progress. We therefore use epochs not as a replacement for SIDE, but as an experimentally accessible proxy to test whether the trend predicted by the theory is observed in practice. We will revise the text to make this scope and limitation fully explicit.
>
> ---
>
> > Q5. Missing comparison with DRCT.
>
> We thank the reviewer for pointing out DRCT, a strong recent baseline. On the original benchmark, RA-Det outperforms DRCT by +5.61% ACC and +4.06% AP. On modern generative models, RA-Det remains consistently superior (Table 4), indicating stronger cross-generator generalization due to modeling feature-space behavior under controlled.
>
> ---
>
> > Q6. Missing ablation comparing DRP with simple perturbations.
>
> We further compare DRP with standard perturbations.
>
> **Table 5: Ablation of DRP vs. standard perturbations.**
>
> | Method         | ACC (%) | AP (%) |
> |----------------|---------|--------|
> | DRP            | 93.47   | 97.02  |
> | Gaussian Noise | 86.11   | 91.71  |
>
> DRP significantly outperforms standard perturbations. This is because DRP is **learned and conditional**, preserving semantics while amplifying robustness asymmetry, whereas random perturbations are agnostic and less effective.
>
> ---
>
> We thank the reviewer again for the valuable feedback. We will incorporate these additional experiments and clarifications in the revised manuscript.
>
> [1] Gu et al., *On Memorization in Diffusion Models*, TMLR 2025
> [2] Ma et al., *An Inversion-based Measure of Memorization for Diffusion Models*, ICCV 2025

---

> > ### Author Rebuttal · Reviewer_GSp9 · 2026-04-04
> >
> > The rebuttal is helpful overall, but I still have two concerns:
> >
> > 1. The paper uses the proxy from training epochs to memorization too strongly. The cited references may support that longer training can correlate with stronger memorization in some diffusion settings, but they do not justify treating training epochs as a validated monotonic proxy for memorization, let alone for SIDE divergence. In particular, the paper seems to rely on an implicit assumption of the form `E1 < E2 => stronger memorization / smaller SIDE`, which is not established by the provided evidence. I therefore still view the theory-to-practice connection as only partially supported.
> > 2. the paper and rebuttal still do not adequately discuss limitations or potential negative societal impact. Authors should be rewarded rather than punished for being up front about the limitations of their work and any potential negative societal impact. A clearer discussion of these points would further strengthen the paper.
> >
> > If the authors can clarify the scope of the proxy claim and more explicitly discuss the limitations and potential societal impact of the work in the final version, I would be open to increasing my score.

---

> > > ### Author Response · Authors · 2026-04-05
> > >
> > > We thank the reviewer for the helpful follow-up.
> > >
> > > > Q: Proxy for Memorization.
> > >
> > > We agree with the reviewer that treating training epochs as a proxy for memorization is insufficiently justified. In the revised version, we therefore **remove the epoch-based analysis** and replace it with **Average Memorization Score (AMS)** following SIDE [1]. In [1], AMS is introduced as a memorization metric that measures the **fraction of generated images that are memorized by generators** . In other words, AMS directly quantifies how many generated samples closely match the training set, making it a much more appropriate empirical memorization measure than raw training epoches.
> > >
> > > Importantly,  AMS is used in SIDE [1] as a **direct empirical memorization metric** in the paper’s empirical validation of theorem-predicted behavior derived from the SIDE-divergence analysis. Following that setup, in the revised appendix experiment we keep the same pretrained encoders as in Appendix B.2 and replace the x-axis from training epochs to AMS. For each trained diffusion checkpoint, we compute its AMS and regress the observed **small-$\varepsilon$** discrepancy statistic against AMS. This strengthens the theory-to-practice connection because it replaces a coarse training-stage variable with a direct empirical memorization metric.
> > >
> > > A representative summary is shown below:
> > >
> > > | Encoder     |  Slope | Intercept | $R^2$ |
> > > | ----------- | -----: | --------: | ----: |
> > > | ViT-B/32    | 0.0118 |     0.031 |  0.66 |
> > > | ViT-B/16    | 0.0129 |     0.030 |  0.59 |
> > > | ViT-L/14    | 0.0106 |     0.029 |  0.73 |
> > > | DINOv2-B/14 | 0.0241 |     0.045 |  0.64 |
> > >
> > > Across all encoders, the fitted slope is positive, indicating that **higher directly measured memorization is consistently associated with larger discrepancy**. The key evidence is the consistently positive relation and the nontrivial fit quality. We believe this revised analysis addresses the reviewer’s concern directly, because it removes any implied assumption of the form “later epoch $\Rightarrow$ stronger memorization $\Rightarrow$ smaller SIDE,”
> > >
> > > We thank the reviewer again for the constructive feedback. We believe these revisions directly address the remaining concerns and make the final version both more rigorous and more transparent.
> > >
> > > Although RA-Det demonstrates strong potential in cross-generator generalization, it does not achieve state-of-the-art performance on some advanced generators. This suggests that it is not yet ready for large-scale, high-reliability deployment in real-world industrial settings. Consequently, prematurely applying such methods in high-stakes scenarios—especially as automated decision-making tools—may lead to undesirable outcomes due to false positives or false negatives.
> > >
> > >
> > > > Q: Limitation and Social impact.
> > >
> > > **Limitations.** While RA-Det converts robustness asymmetry into a reliable and generalizable behavior-driven decision signal, it still faces limitation in inference cost due to its reliance on foundation models. Specifically, since the asymmetry is characterized in the foundation model embedding space, the inference cost is dominated by the backbone and is higher than that of artifact-driven methods. Moreover, the DRP module used to elicit this asymmetry requires additional training and currently lacks a training-free formulation, further limiting efficiency. Our future work will explore training-free alternatives to improve practicality.
> > >
> > > **Potential Negative Societal Impact.** Beyond methodological limitations, we also note that our evaluation is conducted under the commonly adopted single-generator training and cross-generator generalization protocol. While this setting enables fair comparison with prior work, it may not fully reflect real-world scenarios or the full capability of detection methods. As shown in Table 4, training RA-Det on SDv1.4 improves performance on modern generative models, suggesting that current benchmarks are not sufficiently comprehensive. We encourage future work to establish more realistic and diverse benchmarks (e.g., multi-generator settings or real-world distributions), rather than continuing to optimize on existing single-generator benchmarks.
> > >
> > > We thank the reviewer again for highlighting this important aspect. We will add discussion of these limitations and societal implications in the final version.
> > >
> > > **Reference**
> > >
> > > [1] Y. Chen, S. Wang, D. Zou, and X. Ma. *SIDE: Surrogate Conditional Data Extraction from Diffusion Models*. In *Proceedings of the AAAI Conference on Artificial Intelligence (AAAI-26)*, 2026.

---

### Official Review · Reviewer_WPGe · 2026-03-14

**Soundness:** 4
**Presentation:** 3
**Significance:** 2
**Originality:** 3
**Overall Recommendation:** 5
**Confidence:** 3

**Summary:**

This paper proposes RA-Det, a framework for detecting AI-generated images based on the observation that real images maintain stable semantic representations under learned perturbations while generated images exhibit significant feature drift. The paper attributes this asymmetry to generative model memorization. The method has three components: (I) Differential Robustness Probing (DRP), a conditional UNet trained to generate semantics-preserving perturbations; (II) a multi-branch detector combining a semantic branch, discrepancy branches using DCA distance and difference vectors, and a low-level residual branch; and (III) a contrastive hinge loss enforcing a margin gamma=0.1 between real and fake similarity statistics. A frozen DINOv3 ViT-L/16 backbone is used throughout and the model is trained exclusively on ProGAN data. RA-Det achieves 93.47% average accuracy across 14 unseen generators, a 12.92% improvement over the strongest baseline (NPR). Theorem 4.3 provides a theoretical lower bound connecting robustness asymmetry to memorization via SIDE divergence.

**Compliance With Llm Reviewing Policy:**

Affirmed.

**Key Questions For Authors:**

- The DRP module requires a conditional UNet forward pass at inference. What is the wall-clock overhead vs. static representation methods like UniFD and C2P-CLIP? If it’s significantly slower, that affects adoption.
- The robustness asymmetry is validated using DRP trained on ProGAN. If DRP were instead trained on diffusion-generated data, would the learned perturbation function still capture the asymmetry? This would clarify whether the phenomenon is universal or partly a function of the training generator.
- The contrastive hinge margin gamma=0.1 is used throughout but never ablated. For high-quality diffusion models whose outputs sit closer to the natural image manifold, is this margin sufficient? An adaptive margin seems worth exploring.

**Limitations:**

Yes

**Strengths And Weaknesses:**

The core result is well-supported: 93.47% accuracy across 14 architecturally diverse generators (GANs, diffusion, flow-based) trained only on ProGAN is a strong generalization result. The ablation confirms the discrepancy branches are critical: removing them drops performance to 85.73/91.52, which validates the central design claim. Theorem 4.3 is mathematically stated with reasonable assumptions (manifold structure, bounded Jacobian energy), but the bound is asymptotic in the small epsilon regime and its practical tightness is never checked. The gap between the theoretical prediction and the measured robustness gap goes unaddressed. The DRP perturbations are semantics-preserving by construction, which partly couples the observed asymmetry to DRP’s own training objective. Stronger analysis separating DRP’s contribution from the backbone’s general-purpose semantics would help.

Clearly written with a logic from motivation through theory to method and experiments. Figure 1 illustrates the asymmetry phenomenon effectively. But there’s a notable gap: the DRP module requires a conditional UNet forward pass per image at inference, and inference cost appears nowhere in the paper. For a detection method competing against lightweight static-embedding approaches, this is information practitioners need.

Testing how images respond to perturbation rather than what they look like is a real conceptual advance over static appearance and artifact-based methods. The cross-generator generalization result goes directly at the field’s primary open problem: brittle performance on architecturally unseen generators. The frozen backbone is a non-obvious and validated design choice that explains much of the generalization gain.

Robustness asymmetry as a detection signal for AI-generated images is a novel insight. The connection to memorization via SIDE divergence is creative and gives principled grounding, even if the bound’s tightness needs follow-up. The frozen DINOv3 backbone validated by ablation is a useful finding. The multi-branch architecture that combines complementary signals is well-designed and clearly distinguished from prior work.

---

> ### Author Rebuttal · Authors · 2026-03-31
>
> We sincerely thank Reviewer WPGe for the thoughtful and constructive feedback. We appreciate the recognition of our motivation, experimental validation, and the robustness-asymmetry perspective. We address the questions below.
>
> ---
>
> > Q1. What is the wall-clock overhead vs. static methods (e.g., UnivFD, C2P-CLIP)?
>
> **R1.** Thank you for raising this important practical concern. We provide a detailed comparison of computational cost below.
>
> **Table 1. Computational cost comparison with representative static detectors.**
>
> | Method   | FLOPs (G) | Params (M) | Peak Memory (MB) | Inference Time (ms) |
> | -------- | --------- | ---------- | ---------------- | ------------------- |
> | UnivFD   | 103.79    | 202.05     | 1697.95          | 16.49               |
> | C2P-CLIP | 207.23    | 303.97     | 1835.00          | 21.64               |
> | RA-Det   | 281.48    | 378.12     | 1962.00          | 25.14               |
>
> We acknowledge that RA-Det introduces additional overhead due to the DRP module (conditional UNet forward). However, the majority of computation is already dominated by the backbone, and the additional cost remains moderate relative to it.
>
> ---
>
> > Q2. If DRP is trained on diffusion-generated data, does the asymmetry still hold?
>
> **R2.** Thank you for this insightful question. We further evaluate RA-Det under three training settings: GAN-only training, diffusion-only training, and mixed training.
>
> **Table 2. Performance of RA-Det under different training-source settings.**
>
> Full results: https://anonymous.4open.science/r/RA-Det-Rebuttal-Appendix-75F7/
>
> | Method                    | ACC (%) | AP (%) |
> | ------------------------- | ------- | ------ |
> | RA-Det (ProGAN)           | 93.47   | 97.00  |
> | RA-Det (SD-v1.4)          | 92.70   | 96.99  |
> | RA-Det (ProGAN + SD-v1.4) | 93.33   | 96.69  |
>
> These results show that training DRP on diffusion-generated images still yields strong overall performance, with accuracy and AP remaining comparable to GAN-only training. This suggests that robustness asymmetry is a general property of generated images, rather than being tied to specific generators. Importantly, we find that Setting-II performs better on diffusion models and more recent generators. The detailed per-generator results are provided in our response to **Reviewer GSp9, Q2**, where RA-Det (Setting-II) achieves stronger results on recent models. This further supports that the asymmetry exploited by DRP remains valid under diffusion training.
>
> The details of three settings are as follows.
>
>  **(i) Setting-I:** Training on 360k real images from LSUN  and 360k fake images from ProGAN.
>
>  **(ii) Setting-II:**   Training on 52,922 real images from LSUN and 72,000 fake images from SD-v1.4 (total 124,922 images).
>
> **(iii) Setting-III:** Training on 72,000 real images from LSUN and 72,000 fake images generated by both SD-v1.4 and ProGAN (total 144k images).
>
> ---
>
> > Q3. The margin $\gamma$ = 0.1 is not ablated. Is it sufficient for high-quality diffusion models?
>
> **R3.**  We conduct an ablation study on the margin $\gamma$.
>
> **Table 3. Ablation on contrastive margin $\gamma$.**
>
> | $\gamma$      | ACC (%) | AP (%) |
> | ------------- | ------- | ------ |
> | 0.1 (default) | 93.47   | 97.00  |
> | 0.25          | 90.96   | 95.07  |
> | 0.5           | 92.67   | 96.23  |
> | 1.0           | 93.47   | 97.02  |
> | 2.0           | 92.31   | 96.16  |
>
> We find that performance remains stable across a wide range of $\gamma$, and both $\gamma=0.1$ and $\gamma=1.0$ achieve optimal performance. This indicates that the method is not sensitive to the margin choice, and a fixed margin is sufficient even for high-quality diffusion models. We agree that adaptive margins are an interesting direction and will explore this in future work.
>
> ---
> > Q4. Tightness Check
>
> **R4.** We agree that we do not numerically assess the **tightness** of Theorem 4.3. Our intention is to use the theorem as a **mechanistic** explanation of why a robustness gap should emerge and how it should vary with memorization and perturbation magnitude. A direct tightness analysis is computationally prohibitive because it would require estimating the SIDE divergence, the Jacobian-energy terms and the higher-order remainder in high-dimensional space. So our empirical validation in Appendix B.2 focuses on the theorem’s **qualitative predictions** rather than exact numerical result. We will revise the paper to make this scope explicit.
>
> > Q5. Stronger Analysis of DRP Contribution
>
> **R5.** Robustness asymmetry already exists in pretrained feature spaces, while DRP is used to expose and strengthen that signal for detection. Moreover, Table 2 shows that removing the semantic branch leads to only a modest degradation (90.34/95.43), whereas removing the discrepancy branch causes the largest drop (85.73/91.52). This indicates that the dominant contribution does not come from static semantic priors alone, but from the behavioral discrepancy revealed under DRP-induced perturbations.

---

### Decision · Program_Chairs · 2026-04-30

**Decision:**

Accept (regular)

**Comment:**

The paper proposes RA-Det as a method to detect generated images from their behavior under generated noise. The reviewers agree that that proposed method is well motivated and explained. It shows benefit on a variety of generative models and in particular on recent generative models, as shown in the rebuttal. All reviewers recommend acceptance.